# Mitochondrial Dysfunction and Oxidative Stress in Rheumatoid Arthritis

**DOI:** 10.3390/antiox11061151

**Published:** 2022-06-12

**Authors:** María José López-Armada, Jennifer Adriana Fernández-Rodríguez, Francisco Javier Blanco

**Affiliations:** 1Grupo de Investigación en Envejecimiento e Inflamación (ENVEINF), Instituto de Investigación Biomédica de A Coruña (INIBIC), Complexo Hospitalario Universitario de A Coruña (CHUAC), Sergas, 15006 A Coruña, Spain; jenny.fernandez.rodriguez@seergas.es; 2Grupo de Investigación de Reumatología (GIR), Instituto de Investigación Biomédica de A Coruña (INIBIC), Complexo Hospitalario Universitario de A Coruña (CHUAC), Sergas, 15006 A Coruña, Spain; 3Grupo de Investigación de Reumatología y Salud (GIR-S), Departamento de Fisioterapia, Medicina y Ciencias Biomédicas, Facultad de Fisioterapia, Campus de Oza, Universidade da Coruña, 15001 A Coruña, Spain

**Keywords:** rheumatoid arthritis, mitochondria, oxidative stress, metabolism, inflammation, cell death, epigenetic, diet

## Abstract

Control of excessive mitochondrial oxidative stress could provide new targets for both preventive and therapeutic interventions in the treatment of chronic inflammation or any pathology that develops under an inflammatory scenario, such as rheumatoid arthritis (RA). Increasing evidence has demonstrated the role of mitochondrial alterations in autoimmune diseases mainly due to the interplay between metabolism and innate immunity, but also in the modulation of inflammatory response of resident cells, such as synoviocytes. Thus, mitochondrial dysfunction derived from several danger signals could activate tricarboxylic acid (TCA) disruption, thereby favoring a vicious cycle of oxidative/mitochondrial stress. Mitochondrial dysfunction can act through modulating innate immunity via redox-sensitive inflammatory pathways or direct activation of the inflammasome. Besides, mitochondria also have a central role in regulating cell death, which is deeply altered in RA. Additionally, multiple evidence suggests that pathological processes in RA can be shaped by epigenetic mechanisms and that in turn, mitochondria are involved in epigenetic regulation. Finally, we will discuss about the involvement of some dietary components in the onset and progression of RA.

## 1. Introduction

Rheumatoid arthritis (RA) is a prototype of systemic autoimmune inflammatory disorder that is characterized by neovascularization and abnormal synovial hyperplasia associated with local infiltration of several immune and inflammatory cells, which finally damage the structure of the adjacent cartilage and bone driving to functional disability [1]. The development of oxidative stress or redox imbalance result from the excessive level of the different reactive oxygen species (ROS), either by their increased production, the diminishment of antioxidant defenses, or a combination of both. Oxidative stress plays a crucial role in the pathogenesis of RA [2]. In fact, the presence of oxidative stress is associated with clinical parameters of disease activity in RA [3]. Additionally, minor concentrations of antioxidants defenses have been found in serum and synovial fluid of RA patients [4]. Thus, multiple studies have described a shift in the oxidant/antioxidant balance favoring the former in RA serum, synovial tissue and fluid, and contributing to the presence of oxidative damage in cartilage [3,5,6,7,8]. Accordingly, in RA patients, mitochondrial ROS correlates with elevated levels of TNFα in plasma [9] and TNF blocking therapy suppresses oxidative stress and hypoxia-induced mitochondrial mutagenesis as well as recovery of disease evaluated with DAS28 score [10]. In this sense, multiple studies reported the usefulness of drug treatments modulating the RA-related oxidative process [3,10,11,12], but not all studies [13].

Emerging growing evidence has highlighted the rising capabilities of mitochondria in modulating oxidative stress [14]. In this sense, mitochondria are not only recognized as the most important source of reactive oxygen species (ROS), but are also targeted by these molecules [15]. These regulated flares in mitochondrial ROS production plays a central role in the circulation of cellular information; however, in a mitochondrial impairment scenario, there is a boosted production of ROS, which could induce oxidative stress and an inflammatory response [16]. In relation, another key event is the hyper-mutable level of mitochondrial genome mainly as a result of damage caused by the high levels of ROS [17,18], but also by replication errors made by the mtDNA polymerase [18]. Moreover, oxidative damage to mitochondria includes oxidation of proteins, membrane lipids and mtDNA [19], which can also increase mitochondrial destabilization and aggravating oxidative stress, which may lead to further mtDNA mutations leading to a vicious cycle of mitochondrial damaging, thereby causing increased cell damage [20]. This is the reason why mitochondrial impairment has been linked to several human diseases, including RA [21,22,23,24,25,26,27].

Although the past decade has undoubtedly constituted a great revolution in treatment strategies for RA, some patients do no reach low disease activity or become non-responders, and most of the treatments cause long-term adverse side effects [28,29]. In this scenario, dietary bioactive compounds that exhibit anti-oxidant and anti-inflammatory properties have been proposed as potential candidates for the development of new therapeutic interventions [30,31,32,33]. Additionally, multiple evidence suggests that pathological processes in RA can be shaped by epigenetic mechanisms [34]. Thus, it should be noted that nutrition could modify epigenetic and mitochondria are involved in epigenetic regulation [35].

This review will be focused on the importance of mitochondria on oxidative stress and its interplay with metabolism, inflammation and cell death processes. The most recent advances on the existing knowledge concerning the impact of dietary factors on mitochondrial status, and how it may influence the above-mentioned topics in RA course will also be reviewed.

## 2. Physiological Function of Mitochondria and mtROS

Historically, the most recognized function of mitochondria is to be the primary source of cellular energy by coupling the oxidation of fatty acids and pyruvate with the production of high amount of adenosine triphosphate (ATP) by the mitochondrial respiratory chain (MRC) [36]. During mitochondrial oxidative phosphorylation (OXPHOS), the movement of electrons along the first four complexes on MRC produces a proton gradient across the inner mitochondrial membrane that defines the mitochondrial membrane potential (ΔΨm), which is used by complex V to produce ATP. As a result of their activity, mitochondria are the most important source of ROS, generated by the reaction between oxygen and a little quantity of electrons that escape from MRC [14]; but paradoxically, mitochondria are also targets of these molecules [14]. Overall, at the low to moderate physiological levels, the list of targets of redox signaling is lengthy [37]. In this way, ROS is a main signal leading through specific cell signaling pathways (protein phosphorylation, ion channels, and transcription factors), the metabolic regulation and stress responses to support cellular adaptation to a changing environment and stress (immune system, differentiation, cell death, or migration).

In relation to immune response, accumulated evidence has highlighted the vital role of mitochondria as boosters of immunity [3,20,38,39]. Therefore, ROS are the primary host defense tool against infection and adverse scenarios [40]. In this sense, MRC adaptations in immune cells through mitochondrial derived metabolites, redox molecules and naturally mtROS contribute to antimicrobial host defense [41]. In fact, during viral infection, metabolic signals can induce the release of mtDNA from mitochondria with the subsequent type I interferon response [40]. Additionally, principal elements of innate immune essential for fighting pathogen invasion, such as innate immune mitochondrial antiviral signaling protein (MAVS) are modulated by glucose metabolism [42], nucleotide binding oligomerization domain (NOD)-like receptor 1 (NLRX-1) [43] and also mitochondrial dynamics (fusion and fission processes) among others signals [44,45]. For example, dimethylarginine dimethylaminohydrolase 2 (DDAH2) suppresses RIG-I-like receptor (RLR)-MAVS-mediated innate antiviral immunity by stimulating nitric oxide-activated, Drp1-induced mitochondrial fission [45]. Interestingly, some pathogens have acquired the power to negatively regulate mtROS generation and mtDNA sensing, thus potentially suppressing macrophage defense mechanisms, and preventing pathogen clearance and disease resolution [46]. Furthermore, as detailed later, multiple studies pointed out a role for ROS in immune responses through activation of redox-sensitive inflammatory pathways by mitochondrial impairment [22,23] as well as modulation of NLRP3 inflammasome activation [47,48].

Mitochondria, as a remainder of their bacterial origin, possess their own genetic material, which is functionally coordinated with the nuclear genome. Hence, due to the adaptation to the climatic and environmental conditions suffered throughout the different migrations of the human species from its African origin, mitochondria have acquired a series of stable mutations termed haplogroups, which allow them to characterize human populations according to their geographical origin [49]. As a result of their adaptive origin to environmental conditions, each haplogroup presents different metabolic mechanisms and functions. These differences have been related to the predisposition to suffer from different diseases, including rheumatic pathologies such as RA [50]. On the other hand, mtDNA is especially sensitive to mutations caused by ROS due to its proximity to the site of ROS formation as described earlier. The mtDNA encodes 13 polypeptides that are part of the MRC complexes, so mutations in the mitochondrial DNA can compromise the appropriate performance of the mitochondria. In addition, ATP synthase, as well as other MRC enzymes, has been shown to be sensitive to inactivation by oxidative stress [51]. These events can lead to impaired mitochondrial function, which in turn leads to an increase in ROS production, instigating a vicious cycle of mitochondrial collapse.

Finally, the theory of oxidative aging postulates that chronic low-grade inflammation in aging, related to dysregulation of ROS, results in increased damage to organic molecules, which may serve as a bridge between healthy aging and age-related pathological processes [52]. However, a recent study has suggested that oxidative modification target, rather than oxidative modification degree, could underlie ROS-dependent diseases of aging [53]. Undoubtedly, further studies are needed to better determine how redox homeostasis regulates physiology and pathophysiology and to define the basis for novel therapies.

## 3. Interplay between Mitochondrial Oxidative Stress, Metabolic Status and Inflammation in RA

Mitochondria play essential roles at the crossroads of metabolism and innate immunity [40,54]. Relevant to RA, human RA synoviocytes [55,56,57,58] and healthy synovial cells under arthritis-like stimulation, such as inflammatory mediators [59,60] or hypoxia condition [61], impair mitochondrial state of human synovial cells, driving a significant elevation in ΔΨm, shifting their metabolism to aerobic glycolysis, and in turn increasing mutation rates [55,56]. Additionally, a recent study has demonstrated in human synoviocytes that oxidative stress induced by 4-hydroxy-2-nonenal (4-HNE) reprograms energy metabolism reducing enzymatic activity of OXPHOS complexes III and IV and favoring glycolysis, increasing ROS production as well as elevating mitochondrial mutagenesis, contributing to acceleration of inflammatory processes and subsequent defective angiogenesis; thereby favoring a vicious cycle of oxidative/mitochondrial stress [3]. Similar results were obtained in synovial tissue of RA patients [3].

### 3.1. Hypoxia, Oxidative Stress and Inflammation

Oxidative damage in synovial tissue is associated with in vivo hypoxic status in the arthritic joint [7]. In fact, multiple studies have showed that the RA tissue is deeply hypoxic [10,61,62]. Even, a preclinical study showed that hypoxia appears at the pre-arthritic stage and shows colocalization with early synovial inflammation [63]. The transcription factor hypoxia-inducible factor (HIF)-1α is activated in response to low pO_2_ being the key regulator of oxygen homeostasis. During RA, hypoxia induced high quantities of ROS [64] and ROS can also induce the activation of HIF-1 [65,66]. This factor has been found to be highly activated in the RA synovium, being responsible for the expression, among other gene products, of both VEGF (essential in angiogenic processes) and MMPs (essential in angiogenic processes and also in cartilage destruction) [67,68,69]. In addition, it has been related to the induction of COX-2 and IL-8 expression in different cell types [70,71]. HIF-1 is a dimeric factor formed by the HIF-1β subunit, which is expressed constitutively and is maintained at constant levels regardless of oxygen levels; and the HIF-1α subunit, the synthesis of which is constitutive, although its stability is highly regulated by oxygen levels. Under conditions of normoxia, HIF-1α is rapidly degraded; however, under hypoxic environment it stabilizes and translocates to the nucleus where it dimerizes with HIF-1β and the complex formed becomes transcriptionally active [72,73]. The activation of HIF-1α is key for the adaptation of cells to hypoxic conditions, activating the transcription of genes related to angiogenesis or the glycolytic pathway in an attempt to restore the supply of oxygen and correct the defect in the generation of ATP [74]. Other factors besides hypoxia and ROS can induce the activation of HIF-1, such as the cytokines IL-1β and TNFα, hormones, growth factors or mechanical stress, all of them important factors in the pathology of RA [75]. Thus, IL-17 and TNFα combination induces a HIF-1α-dependent invasive phenotype in synoviocytes [76]. HIF-1α also is essential for myeloid cell-mediated inflammation as well as T_H_1/T_H_2 lymphocytes ratio in animal model of arthritis [77,78]. Hypoxia also activates the expression of NF-κBp65 through the canonical signaling pathway [79], and the STAT3 signaling [71]. Finally, a more recent in vitro study with RA synoviocytes shows that mitochondrial dysfunction and increased glycolysis induced by hypoxia was associated in vivo with synovial expression of several glycolytic enzymes as well as glucose transporter 1 (GLUT1) to improve uptake of glucose in RA patients with low pO_2_ levels [80]. Additionally, Tannahill et al. show that succinate induces IL-1β secretion through HIF-1α induction in macrophages [81].

To the best of our knowledge, it has also been reported that hypoxia can induce Notch activation in RA [82]. The Notch signaling pathway is a key developmental route that regulates many cellular processes, including proliferation, cell survival/death and differentiation through intracellular signal transmission that involve receptor-ligand interactions between adjacent cells [83]. In this sense, the activation of Notch signaling is involved in lymphocytes, synoviocytes and endothelial cells of RA patients [82,84,85,86] being promoted by TNF in RA synoviocytes [87]. Additionally, several studies have shown that Notch activation accelerates the production of proinflammatory mediators in RA [82]. On the other hand, Notch-1 and Notch-3 mediate hypoxia-induced synovial fibroblasts activation and angiogenesis in RA [82]. In relation to T helper cells from RA patients, a significantly altered expression profile of Notch receptors and enhanced activation of Notch signaling is displayed compared with healthy controls [85]. Additionally, Notch-regulated miR-223 targets the aryl hydrocarbon receptor pathway and increases cytokine production in macrophages from RA patients [88]. In this regard, the amelioration of experimental arthritis by silencing miR-223 has been described [89] and, in addition, miR-146a modulates macrophage polarization by inhibiting Notch-1 pathway in macrophages [90]. Finally, inhibition of Notch signaling ameliorates experimental inflammatory arthritis [91,92,93]. Thus, Notch signaling could be a potential pharmacological target for RA treatment.

As it was commented above, multiple evidence suggests that pathological processes in RA can be shaped by epigenetic mechanisms [34,94]. In this sense, it should be noted that hypoxia is one of the best-described elements influencing the epigenotype [95]. Therefore, advances in understanding how hypoxic RA tissue induces epigenetic modulations may provide potential advances for novel therapies.

### 3.2. Oxidative Stress, Cell Metabolism and Inflammation

Overall, a healthy synovium has a very different metabolic degree compared to RA synovium (pannus), since one of the main signatures of the RA synovial tissue is a remodeled cell metabolism. Indeed, several type of RA cells (synoviocytes, macrophages, CD4+ T cells, T-helper type 17 (T_H_17) cells and dendritic cells) choose the glycolytic route to produce ATP instead of the more productive OXPHOS route [57,96,97,98,99,100,101,102,103]. This switch in cell metabolism, not related to oxygen availability, is the so-called Warburg effect or aerobic glycolysis [104]. In this sense, recent findings point out glycolysis rate-limiting enzymes as novel potential regulators of RA pathogenesis [105]. In fact, it has been shown that the expression of the first rate-limiting enzyme of glycolysis, hexokinase 2 (HK2), is elevated in the RA synovial tissue, as well as that its overexpression in the synovial lining of a preclinical model promotes hypertrophy of healthy synovium and an aggressive synovial phenotype. By contrast, HK2 deletion in FLS decreases its invasive phenotype, both in vitro and in a model of arthritis [57]. The second key rate-limiting enzyme in the glycolytic pathway is phosphofructo-2-kinase/fructose-2, 6-bisphosphotase (PFKFB). PFKFB is overexpressed in RA synoviocytes regulating the expression of multiple inflammatory cytokines and chemokines, promoting cell proliferation, invasion and migration, and concomitant pannus formation [106]. Additionally, RA CD14^+^ monocytes display increased levels of key glycolytic enzymes HK2 and PFKFB, and demonstrate a reliance on glucose consumption and inflammatory dysfunction, a phenotype that precedes clinical manifestation of disease [107]. By contrast, in naïve CD4+ T cell of RA patients show low levels of PFKFB and high levels of glucose-6-phosphate dehydrogenase (G6PD) inducing a decrease in glycolytic pathway and an increase in pentose phosphate pathway that lead to ROS exhaustion and low ATP levels [108,109]. Finally, pyruvate kinase M2 (PKM2), a critical enzyme that catalyzes the last step of glycolysis, is usually upregulated in proliferative cells. Thus, RA synoviocytes present high PKM2 expression according to increased glycolytic activity [98]. In relation, PKM2 knockdown suppressed migration, invasion, and the expression of IL-1β, IL-6, and IL-8 by TNFα–treated RA FLSs [110]. Additionally, metabolic reprogramming of macrophages instigates CCL21-induced arthritis [111]. As a result of all these changes, RA synovial fluid and synovial tissue have a significant elevation in lactate and a decrease in glucose levels, which correlated with the marker of disease activity PCR [112,113]. Of note, pro-inflammatory profile of autoimmune CD8+ T cells relies on increased lactate dehydrogenase A (LDHA) activity and aerobic glycolysis [114]. Additionally, lactate boosts the switch of CD4+ T cells to an IL-17+ subset [113]. Besides, lactate together with other glycolytic intermediates could contribute to RA pathogenesis through potent stimulation of angiogenesis [80]. Finally, it should be remarked that recent studies suggest that the hypoxia-lactate axis could temper inflammation [115]. Therefore, lactate could counterbalance the inflammatory scenario triggered by hypoxic environment, promoting a metabolic switch from inflammatory macrophage to homeostatic M2 like-polarization by epigenetic modifications (histone lactylation) [95].

Other intermediates of mitochondrial tricarboxylic acid (TCA), such as succinate, fumarate and citrate, are relevant in RA pathogenesis. Succinate is abundant in synovial fluids from RA patients. Furthermore, succinate promotes the stabilization of HIF-1α and enhances pro-inflammatory IL-1β production in a succinate receptor (GPR91)-dependent manner [116,117]. Thus, succinate receptor deficiency attenuates arthritis by reducing dendritic cell traffic and expansion of TH17 cells in the lymph nodes [118]. Succinate can also induces synovial angiogenesis in RA through metabolic remodeling and HIF-1α/VEGF axis [67]. With regard to citrate, which can be metabolized into itaconate by the enzyme immune responsive gene-1 (IRG1), fumarate has been observed to mediate anti-inflammatory effects [119,120]. In fact, increasing plasma itaconate levels in early RA patients correlated with improved DAS44 score and decreasing levels of C-reactive protein (CRP) associated macrophage activation [121]. Additionally, activation of Nrf2/heme oxygenase (HO)-1 signaling pathway by dimethyl fumarate ameliorates complete Freund’s adjuvant-induced arthritis in rats [122].

Notably, several drugs currently in use for the treatment of RA could exert a number of anti-inflammatory actions by affecting metabolic signaling pathways [80,119,121]. Thus, tofacitinib, an oral Janus kinase inhibitor for the treatment of RA, decreases in RA synoviocytes [100] and CD8+ T cells [114] the mRNA expression of different glycolysis rate-limiting enzymes such as the mentioned HK2, PFKFB or other molecules involved in aerobic glycolysis, as GLUT1, also higher in RA synoviocytes, which in turn results in the attenuation of inflammatory response and hyperplasia state. In the same line, glucose-lowering agents such as metformin or thiazolidinediones (TZDs) have shown potential anti-inflammatory activities and protective effects on RA [123,124,125]. Interestingly, blockade of glycolysis alleviates inflammatory phenotype in RA macrophages and RA fibroblasts, even when metabolic regulation of both cell types is distinct [126].

Notably, a recent study described different metabolic phenotype of synoviocytes that discriminates acute, self-limiting synovitis (resolving arthritis) from a persistent very early RA (veRA) [127], which may in turn explain differences in phenotypes between subgroups of patients. veRA synoviocytes may exhibit a delay in upregulating glycolysis in response to inflammatory mediators. Intriguingly, this later study not showed differences between both metabolic phenotypes relative to glucose and lactate levels. Definitely, it would need further investigation about the role of glucose metabolism in the pathogenesis of RA to explore novel therapeutic strategies as well as new panels of biomarkers to help to understand the heterogeneity of RA.

### 3.3. Oxidative Stress, Mitochondrial Dysfunction and Inflammatory Response

There is accumulating evidence for an apparent connection between oxidative stress, mitochondrial dysfunction and inflammatory and immune responses [20,24,26,54,128]. Damaged/dysfunctional mitochondria could release damage-associated molecular patterns (DAMPs, also known as alarmins) such as mtROS and also mtDNA. DAMPs can act through modulating innate immunity via redox sensitive inflammatory pathways (i.e., NF-κB or AP-1) or direct activation of the inflammasomes, cytosolic receptors that once activated induce the maturation of the pro-inflammatory cytokines IL-1β and IL-18 through caspase-1 activation [47,48,129]. NLRP3 is one of the most extensively described inflammasome receptors for its relevant role in the pathogenesis of many sterile inflammatory diseases such as RA. Redox sensitive inflammatory pathways as well as inflammasome pathway are elevated in RA [48,130]. Remarkably, both pathways could interact between them boosting the inflammatory response. In this line, it has also been described mitochondrial impairment could sensitize cells, causing a significant exacerbation of cytokine-induced inflammatory response through ROS generation and sensitive-redox transcription factor as NF-κB [22,23,131]. Specifically, when mitochondrial impairment was induced in normal synoviocytes by commonly mitochondrial inhibitors, an otherwise less efficient concentration of IL-1β was as effective as a 10 times greater concentration of IL-1β in the absence of mitochondrial impairment [22]. On the other hand, several studies and our own research have demonstrated in vitro that several inflammatory mediators, such as the cytokines TNFα and IL-1β, as well as reactive RNS may induce mitochondrial alterations [27,132,133]. Parallel, oxidative stress increases mtDNA mutations and ROS [3], contributing to perpetuate a vicious cycle of oxidative/mitochondrial stress. The function of NLRP3 is monitoring the cytosol for stressful environments. NLRP3 is activated after exposure to a wide range of stimuli including those of mitochondrial origin (DAMPs) such as ATP, ROS, cardiolipin or oxidized mitochondrial DNA fragments and other stimuli such as nigericin, bacterial toxins such as lipopolysaccharide LPS, viruses or alcohol [134]. Activation of typical NLRP3 requires two signals: a first signal in which transcription of the pro-IL-1β and pro-IL-18 genes is induced, as well as NLRP3, which dependent on the activation of the NF-κB signaling pathway (cytokine or Toll-like receptors). Moreover, a second signal, in which the extra- or intra-cellular stimuli recruit the adapter molecule caspase recruitment domain (ASC) and caspase-1 to assembly NLRP3 inflammasome, leading to caspase-1 activation and, consequently the maturation of IL-1β and IL-18. Thus, the pivotal roles of mitochondria in the initiation and regulation of NLRP3 is beyond any doubt. Curiously, NF-κB via autophagy inhibits NLRP3 inflammasome activation through p-62-dependent clearance of damaged mitochondria [135]. By contrast, extrinsic and intrinsic apoptosis activate pannexin-1 to drive NLRP3 inflammasome assembly [136], and caspase-1 initiates apoptosis in the absence of gasdermin D [137]. Finally, note that epigenetic dynamics is a key regulator of the expression of inflammasome components and its further activation [138]. Therefore, targeting mitochondrial maintenance may control cell homeostasis, and in turn, delay aging and prevent related diseases such as RA.

#### 3.3.1. Redox-Sensitive Inflammatory Pathways in RA

There is an extensive variety of cellular redox-sensitive signaling processes such as the activation of nuclear factor-kB (NF-κB), the activator protein-1 (AP-1) that lead to a transcriptional up-regulation of a number of genes involved in inflammation and/or fibrogenesis [139] or the nuclear factor erythroid-2-related factor 2 (Nrf2), which is a crucial transcription factor resolving inflammatory and apoptosis process [140]. Since mitochondria are the major source of ROS, mitochondria are also critical elements in the control of cellular redox-sensitive signaling.

In RA, the transcription factor NF-κB is highly activated and represents one of the main inflammatory mediators since it is involved in the induction of numerous pro-inflammatory cytokines such as IL-1β, IL-6, IL-8 or TNF-α in monocytes, macrophages and also in synoviocytes [22,23,141]. In turn, these cytokines are capable of activating the NF-κB factor in other immune cells such as T and B lymphocytes and in synoviocytes, thus inducing the expression of additional inflammatory cytokines and chemokines, which leads to an inflammatory loop, with a greater recruitment of inflammatory cells from the immune system and the spread of inflammation [142]. In relation, oxidative stress could modulate the cytokine response of differentiated Th17 and Th1 cells during RA [143]. Another important role of NF-κB in RA is to promote synovial hyperplasia by promoting cell proliferation and inhibiting apoptosis, assuming a link between the inflammatory process and the reduced rate of cell death [144]. In addition, it contributes to tissue destruction since it is involved in the synthesis of MMPs and vasoendothelial growth factors such as vascular endothelial growth factor (VEGF), which favors, to a large extent, the characteristic invasive capacity of the AR synovium. Surprisingly, it has been demonstrated that mitochondrial dysfunction may generate low-grade inflammatory and matrix degradation via mitochondrial calcium exchange, ROS production and NF-κB activation [22,23,145]. The aforementioned hypoxia also activates the expression of NF-κB through the canonical signaling pathway [79], displaying synergistic behavior during hypoxic inflammation and contributing to stablish an inflammatory loop [146]. Additionally, remember again that NF-κB could restrict inflammasome activation via elimination of damaged mitochondria providing an essential regulatory loop to limit its own inflammatory response [135]. Interestingly, NF-κB polymorphism are also important in the pathophysiology of RA. Thus, a recent study associate NFKB2 polymorphisms with the risk of developing RA and the response to TNF inhibitors [147]. In relation, it has been demonstrated that the combination of NF-κB targeted siRNA and MTX in a hybrid nanocarrier could effectively treat the RA in a preclinical model avoiding the adverse effects of MTX and opening a novel therapeutic approach in the treatment of RA [148]. In this way, targeting NF-κB activation could represent a novel therapeutic approach to attenuate RA development.

ROS also activates the AP-1 pathway followed by expression of pro-inflammatory genes such as TNFα and MMPs [149,150]. Therefore, some of the effects of AP-1 activation are the increase in cell proliferation, angiogenesis, induction of osteoclastogenesis or matrix degradation by synoviocytes [151,152,153]. Additionally, a recent study has identified new functionally relevant mutations in the coding regions of the human Fos and Jun proto-oncogenes in RA synovial tissue [154]. Finally, and relative to progress toward prevention, the therapeutic potential of a novel selective histone deacetylase 6 inhibitor in a murine model of arthritis via blockade of NF-κB and also AP-1 activation has been described [155].

Nrf2 regulates the expression of more than 200 genes involved in antioxidant defense [5]. It is activated in the synovium of RA patients and increased levels of its target enzyme HO-1 have also been found in RA synovial fluid. Thus, although this antioxidant factor is unable to control the oxidative stress associated with the pathology, its silencing would cause an aggravation of the symptoms and joint destruction [8]. On the contrary, its induction with different agents attenuates the severity of arthritis. Furthermore, the Nrf2 inducer sulforaphane has also shown the ability to induce apoptosis in TNF-stimulated synoviocytes, but not in healthy synoviocytes [156]. More recently, the anti-arthritis effect of sulforaphane has also been associated with the inhibition of both B cell differentiation and the production of inflammatory cytokines [157]. As commented above, the endogenous mitochondrial TCA metabolite itaconate is increasing in plasma from early RA patients and correlate with improved DAS44 score and decreasing levels of C-reactive protein (CRP) [121]. In relation, it has been demonstrated that itaconate activates Nrf2 via alkylation of its inhibitory protein Kelch-like ECH-associated protein 1 (KEAP1) [120]. Remarkably, Nrf2 exerts its anti-inflammatory effects, at least in part, through the inhibition of NF-κB pathway. In this sense, there is a cross-talk between Nrf2 and NF-κB pathways [5].

Further studies are necessary to establish effective strategies to maintain the appropriate levels of oxidative stress and to offer novel potential therapeutic strategy for the treatment of RA.

#### 3.3.2. NLRP3 Inflammasome in RA

Compelling evidence has described that NLRP3 inflammasome, and in turn IL-1β secretion, are highly activated in both RA patients and arthritis preclinical models in several type of RA cells (macrophages or monocytes, CD4^+^T cells, Th17 and synoviocytes) [48,158,159,160,161]. Therefore, monocytes from RA patients display an increased production of IL-1β via NLRP3 inflammasome [161]. Notably, NLRP3 inflammasome regulates Th17 differentiation in RA [162] as well as Th2 differentiation [163]. Besides, Th1 immunity requires complement-driven NLRP3 inflammasome activity in CD4^+^ cells [164]. These results suggest that NLRP3 is also involved in adaptive immunity and not just innate immunity. On the other hand, drugs used in chronic arthropathies [165], but not in all studies, modulate inflammasome activation on RA. Thus, tofacitinib restores the balance of γδTreg/γδT17 cells in RA by inhibiting the NLRP3 inflammasome [166]. However, it has been described that, paradoxically, glucocorticoids can also have proinflammatory influence on the immune system through upregulation of the NLRP3 inflammasome [167]. Remarkably, it has been observed how a genetic modification that predisposes to the development of arthritis in mice results in greater activation of the NLRP3 inflammasome complex and likewise, the deletion of a functional NLRP3 complex produces notable improvements in the disease [168]. In this line, Guo C. et al. have recently shown an increased expression of NLRP3, caspase-1 and IL-1β in the synovial tissue of mice in which arthritis was induced, and which was reduced by treating animals with the selective inhibitor of NLRP3, MCC950 [48]; however, phase II clinical trials in RA where inhibitor MCC950 was used were interrupted because of its hepatoxicity [129]. By contrast, caspase-1 mediated IL-18 activation in neutrophils promotes the activity of RA in a NLRP3 inflammasome independent manner [169]. Interestingly, a recent study has described how anti-citrullinated protein antibodies (ACPAs) promote IL-1β production in RA by activating the NLRP3 inflammasome, suggesting a new role of ACPAs in RA pathogenesis [170]. Moreover, NLRP3-mediated IL-1β secretion could be regulated by a variant within the gene locus encoding PTPN22, which has emerged as an important risk factor for auto-inflammatory disorders, including RA [171]. In addition to this, polymorphisms and expression of inflammasome genes are associated with susceptibility, disease activity and anti-TNF treatment response in RA [172,173,174], while it has also been described how several microRNA (miR-33, miR-20a and miR-30a) act as positive regulator of the NLRP3 inflammasome in RA macrophages [175,176,177]. More recently, the synergy of complement C1q with PTX3 in promoting NLRP3 inflammasome over-activation and pyroptosis in RA has been detailed [178]. In relation, the DNA repair nuclease MRE11A works as a mitochondrial protector and prevents T cell RA pyroptosis and tissue inflammation [179]. Finally, a last study by Jäger et al. reports how calcium-sensing receptor-mediated NLRP3 inflammasome response to calciprotein particles drives inflammation in RA monocytes and enhances inflammatory arthritis and systemic inflammation [180]. Collectively, these results suggest that direct therapies targeting NLRP3 inflammasome could be a potential therapeutic strategy for the treatment of RA [47].

## 4. Mitochondrial Oxidative Stress and Cell Death in RA

An altered ratio of programmed cell death could actively contribute to the development of synovitis and finally, the formation of the synovial *pannus*. Therefore, synovial hyperplasia could be the consequence of the dysregulation of programmed cell death in various associated cell types (synoviocytes, T cells, B cells, monocyte-macrophages and neutrophils) in a context of excessive cell proliferation, such as that observed in RA synovium [181].

Cell death is a physiological process involved in maintaining tissue homeostasis during normal development of different structures, natural aging process or in case of injury; and acts by establishing a balance between the number of new cells and those that are damaged. In this way, defective cell death can lead to a pathological scenario [26,182]. This way, uncontrolled cell proliferation results in the development of diseases such as cancer or RA, while an excessive level of cell death can lead to diseases such as Alzheimer’s or Parkinson’s [182]. Mitochondria also have a central position in regulating cell death [182,183]. As a result, it is not conspicuous that damaged mitochondria have been associated to multiple acute and chronic diseases as RA [184].

Depending on the morphological alterations of dying cells, the mechanisms by which these and their fragments are eliminated, classically cell death has been established in three types: type I cell death or apoptosis, type II cell death or autophagy and type III cell death or necrosis [185]. However, the last Cell Death Nomenclature Committee of 2018 proposes an updated and complete classification, focused on morphological, molecular and functional aspects of the cell death process. Thus, up to 12 different types of cell death have been identified [185] that can be executed following different or overlapping signaling pathways, even sharing the same molecular process [185]. Additionally, regulated cell death has long been considered as an immunologically silent or even tolerogenic event [186]; however, depending of the cell death type, dying cells release and exhibit different signals at their surface, which could dictate the immunogenicity of cell death [136,187,188,189,190].

Apoptosis or type I programmed cell death is the most common form of cell death [188]. Morphologically, it exhibits a series of highly organized alterations that include cytoplasmic contraction, chromatin condensation, nuclear fragmentation, and vesicle formation in the plasma membrane [188]. All of these events culminate in the formation of small vesicles known as “apoptotic bodies” that enclose the intracellular components [185]. These compartments can be easily captured by neighboring cells with phagocytic activity and degraded within the lysosomes [185], which means that the frequency of apoptosis is often undervalue. To date, two main signaling pathways have been described that can trigger the execution of apoptotic cell death: the extrinsic pathway, which involves the classical interaction between ligand and receptor (also called death receptor) on the cellular surface such as TNFα receptor-1 (TNFR1) or FAS; and the intrinsic or mitochondrial pathway [185]. Both signaling pathways involve the activation of a family of enzymes called caspases [191]. Although the two routes of apoptosis are very different, both have in common a last phase of execution of cell death that is characterized for being irreversible and for involving the activation of effector caspases-3 and -7 [192,193]. Interestingly, caspase-1 is able to initiate apoptosis [137], highlighting the interplay between cell death and inflammasome activation [194]. As commented above, the intrinsic pathway is associated to mitochondria and is inducible by multiple stimuli, including high levels of mtROS [183,195]; these produce the activation of BH3 proteins that, in turn, activate BAX and BAK, triggering the permeabilization of the mitochondrial outer membrane [194]. In fact, the crucial event of intrinsic apoptosis is the mitochondrial outer membrane permeabilization, which leads to the liberate of cytochrome *c* and the assembling of the prominent complex apoptosome [196,197]; that subsequently causes the downstream cascade activation. Mitochondrial inner membrane permeabilization enables mtDNA release during apoptosis [54,190] and underpins various inflammatory pathways triggered by mtDNA efflux.

Autophagy or type II programmed cell death was originally identified as a strategy for cell survival during lack of nutrients or other stress situations, but it has been observed that it is capable of mediating cell death depending on the context in which it is found. In this way, autophagy plays a delicate role in the regulation of cell survival and death. Cell death by autophagy is the result of excessive induction of the autophagic process because of intracellular damage or an excessively high number of non-functional organelles; so once it is pushed to the limit and has reached the “*point of no return*”, it ends with the death of the cell. This is morphologically characterized by the accumulation of cytoplasmic vesicles (destruction of large amounts of cytoplasm occurs) resulting in irreversible cellular atrophy and the consequent collapse of crucial cellular functions [185,198]. This type of death is a regulated and catabolic process dependent on lysosomal action, which therefore makes it easier for cells to eliminate cellular components that are damaged or have ceased to be functional (mitochondria, endoplasmic reticulum, peroxisomes), proteins that are poorly folded or pathogenic, in order to preserve cellular homeostasis [185]. Cell death by autophagy is caspase-independent. The mTORC1 complex is the major sensor of autophagy, since it avoids the initiation of the autophagy by Atg1 (autophagy related 1). In relation, Atg genes control the autophagy development. Autophagy is classically organized in five stages. Initiation of the autophagic process with the formation of the ULK1 complex. Phagophore formation and nucleation in which beclin-1 is required. Elongation with the recruitment of processed LC3 to the membrane of the growing phagophore and interaction of the p62/SQSTM1 adapter with cytosolic cargo. Maturation in which the phagophore encloses the cellular cargo, giving rise to the mature double-membrane autophagosome. Finally, the autophagosome and lysosome fusion to form the autophagolysosome where the degradation of the cytosolic cargo occurs by lysosomal enzymes proteolytic degradation. It is important underline that several authors have described a *non-canonical* pathway of autophagy. This term is used to distinguish a process in which autophagosome formation and maturation is independent of beclin-1 [199]. Mitochondria could act regulating autophagy through ROS as well as by interacting with lysosomes and endoplasmic reticulum [200]. Apoptosis and autophagy were studied for a long time as two totally isolated mechanisms that represented two mutually exclusive cellular states; instead, the steady accumulation of evidence over the past decade has suggested that these two modes of programmed cell death may often be interconnected by complex networks of proteins [201]. Thus, depending on the context surrounding the cell, the two main cell death pathways cooperate in a balanced interaction that promotes cell survival or cell death [202,203]. Therefore, mitochondria have plausible therapeutic options for the treatment of many diseases [182,183].

### 4.1. Apoptosis in the Pathophysiology of RA

RA synoviocytes undergo fundamental changes during the course of the disease, adopting an activated *tumor-like* phenotype associated with the acquisition of resistance to apoptosis [184]. In this way, insensitivity to apoptosis leads to abnormal proliferation of RA synoviocytes [181]. Numerous events that converge in the RA synovial environment promote the survival of synoviocytes and hinder their elimination through apoptosis. Thus, two of the most significant alterations in T-cell impaired in RA is the permanent activation of T-cells and the subsequent abnormal proliferation state which also stimulate the proliferation of fibroblasts within the joint synovial tissue contributing and favoring, to a great extent, the survival of active synoviocytes [204]. Likewise, it has been shown how the cells present in the RA synovium show an increased expression of the antiapoptotic proteins of the Bcl-2 family (Bcl-2 and Mcl-1) that would act by restricting the susceptibility to the intrinsic pathway of apoptosis [205]. In agreement, it has been observed in in vitro studies with RA synoviocytes that the stimulation of these cells with proinflammatory mediators such as TNF-α or IL-1β increases the expression of Bcl-2 and, therefore, protects them from cell death in an inflammatory environment. In addition, sometimes this expression is located in lymphoid aggregates, which could suggest a protective mechanism of T and B cells against cell death [205]. In this line, the survival of T lymphocytes ensures that both non-immune cells, such as synoviocytes, as well as immune cells, including B cells, macrophages, dendritic cells, mast cells or neutrophils, continue to survive in the RA synovium where they perpetuate the chronic inflammatory process of RA [204,206]. Interestingly, the inhibition of NF-κB signaling pathway induces apoptosis and suppresses proliferation and angiogenesis of human fibroblast-like synovial cells in rheumatoid arthritis [207]. Additionally, it has been described as hypoxia reduced RA synovial fibroblast cell apoptosis through Notch-3, whereas an increase in autophagy bodies under hypoxia can be limited by siNotch-3 [82].

On the other hand, recent studies identified a new crosstalk between metabolic status, mitochondria and apoptosis in RA. Thus, moderate extracellular acidification inhibits capsaicin-induced apoptosis through regulating calcium mobilization, NF-κB translocation and ROS production in synoviocytes [208]. Moreover, the glycolytic enzyme HK2, which is elevated in RA synoviocytes can bind to mitochondrial membrane via its interaction with the outer membrane porin protein voltage-dependent anion channel (VDAC) and inhibit the release of cytochrome *c*, and in turn might protect synoviocytes from apoptosis [57]. In relation, commonly used drugs for the treatment of RA could exert their therapeutic benefits, at least in part, by regulating apoptosis [181]. As an example, a JAK inhibitor suppressed the pro-inflammatory behavior of RA synoviocytes accelerating apoptosis and abrogating thickening of the synovium [209].

### 4.2. Autophagy in the Pathophysiology of RA

What role does autophagy play in RA? Similar to other autoimmune diseases, autophagy plays a dual role in RA [26,210], showing both a therapeutic and a pathogenic effect [211,212]. In addition to the defective apoptosis in resident synoviocytes as well as in immune and inflammatory cells which infiltrate the RA synovium and that contribute to the persistence of RA, it has been described how the autophagy level in the synovial tissue of patients with active RA are greatly increased, correlating with disease severity [213]. Furthermore, increased autophagy in CD4^+^ T cells results in T-cell hyperactivation and also contributes to their apoptosis resistance [206]. In relation, low oxygen levels in the RA joint are inversely associated with an increase in autophagy, and consequently, in synovial inflammation and oxidative damage, given the implication of this type of cell death in the activation of immune function [214]. In addition, autophagy increases auto-antigen presentation by antigen presenting cells [215]. However, other studies have suggested that severe endoplasmic reticulum stress in RA synoviocytes leads to cell death through the formation of autophagic vesicles. In particular, Kato M. et al. explain that: “in contrast to the apoptosis-resistant phenotype of the AR synoviocytes, a potential Achilles’ heel has been identified in these cells by inducing cell death by autophagy” [210]. In addition, it has been described that a combination of the mTOR (suppressor of autophagy) inhibitor everolimus and MTX may have clinical benefit for the therapy of RA in patients who have an insufficient response to MTX monotherapy [216]. In addition, there is no doubt that dysregulated autophagy could modulate inflammasome activity and be a main driver of multiple autoinflammatory and auto-immune diseases [131,135,217]. In fact, autophagy inhibits NLRP3 inflammasome activation through p-62-dependent clearance of damaged mitochondria [135] and, subsequently, regulating innate immune responses by inhibiting the release of mitochondrial DNA mediated by the NLRP3 inflammasome [131]. In this sense, our group has recently described that autophagy activation by resveratrol reduces the severity of experimental RA that pharmacological intensification of autophagic flux by resveratrol in an RA preclinical model limits the cross-talk existence with inflammation [218,219]. In agreement, autophagy enforces functional integrity of regulatory T cells by coupling environmental signals and metabolic homeostasis [220]. In fact, PFKFB deficiency impairs ATP generation, autophagy, and redox balance in RA T cells with increased susceptibility to apoptosis [108]. Finally, a recent study showed how RA T cells with hyperactivated mTORC1 pathway leads to pro-inflammatory T_H_1 and T_H_17 helper T cells and promotes synovial tissue inflammation [221]. Due to the central role of mitochondria in cell death, understanding of how mitochondria fine-tune the interplay between metabolism, apoptosis, and autophagy might provide most effective strategies for clinical treatment, also in RA.

## 5. Interplay between Mitochondrial and Epigenetic Mechanisms in RA

Epigenetic concerns to changes in the expression and functions of genes that are heritable even when the DNA sequences persist without changes. Thus, epigenetic changes include DNA methylation, histone modification, and expression of micro-RNAs [222]. Multiple emerging evidences describe differences in the epigenome of disease-relevant cells in RA patients in relation to healthy subjects [223,224,225]. In fact, DNA from peripheral blood mononuclear cells (PBMCs) and RA synoviocytes are hypomethylated in active RA and methylation correlates with disease activity [94,226]. Interestingly, it should be noted that the hypoxic status that defines the arthritic joint is one of the best-defined features governing the epigenotype [95,227]. On this point, lactate could compensate the inflammatory setting triggered by hypoxic condition, promoting a metabolic switch from inflammatory macrophage to homeostatic M2 like-polarization by epigenetic modifications (histone lactylation) [95]. Methylation also promotes inflammation and activation of fibroblast-like synoviocytes in RA [228,229] as well as is associated with the level of transcript in T lymphocytes from RA patients [230]. Notably, differential DNA methylation correlates with response to methotrexate in RA PBMCs [231,232]. In fact, the key drivers of RA synovitis, TNF and IL-1, reprogram the epigenomic landscape of FLS by altering chromatin and contributing then to induce unremitting expression of arthritogenic genes in RA synoviocytes [233,234,235]. Additionally, note as different epigenetic changes are involved into NLRP3 inflammasome activation [138]. Histone modification and micro-RNA are too involved in epigenetic regulation in the pathogenesis of RA. Thus, a reduced activity of HDAC3 and increased acetylation of histones H3 has been described in PBMCs of RA patients [236]. In relation, a recent study showed in a preclinical model of RA as well as in human RA synoviocytes the therapeutic potential of a novel selective histone deacetylase inhibitor [155,237]. As an example of micro-RNA-mediated pathogenesis of RA, plasma micro-RNA-22 is associated with disease activity in well-established RA [238]. Besides, there is a lower circulating of miR155 levels in RA patients compared to healthy controls that interestingly correlated with a miR-155 gene methylation level significantly higher in RA patients [239]. 

For years, it has been known that nutrition plastically modulates the epigenetic landscape, but also epigenetic marks exert profound effects on metabolic genes regulation favoring the choice of the glycolytic route to produce ATP instead of the more productive OXPHOS route [35]. In this context, mitochondrial function/dysfunction is the main actor and for this reason, the interplay between metabolism and epigenetic changes and mutations in mitochondrial genetics is a novel subject of interest [35,127]. Thus, mitochondria could be involved in epigenetic modulation through bidirectional crosstalk between mitochondrial and nuclear genome. Besides, mitochondria genome methylation could be also involved in the process. In this regard, the metabolic intermediate of mitochondrial TCA, succinate has been related to changes in DNA methylation and associated histone proteins, which in turn regulate gene expression [240]. In relation, Torres et al. described the switch in the epigenetic environment of genes associated to the regulation of nutrient transporters in RA synoviocytes [241].

Although still in the midway between facts and fiction, the reversibility of epigenetic marks, and in turn, the possibility of epigenetic reprogramming could open novel pharmacologic opportunities in the treatment of RA [34]. Moreover, epigenetic signatures could provide new panels of biomarkers to offer a new perspective for the diagnosis of RA.

## 6. Dietary Factors on Mitochondrial Status in RA

Accumulating evidence has identified some dietary factors as important risks factors for RA. In this sense, conducting a literature search on studies that analyze the impact of obesity on disease activity and treatment response in RA, there is no doubt that obesity is associated with more severe symptoms among RA patients. However, it remains unclear whether poorer treatment response rates are related to reduced efficacy of therapies [242]. Obese patients may have increased levels of inflammatory cytokines and adipokines [243,244]. In addition, insulin resistance is closely associated with an increased risk of subclinical atherosclerosis in patients with rheumatoid arthritis (RA). Anti-TNF therapy reduces insulin resistance and improves insulin sensitivity in patients with severe RA. New findings have shown, however, that the efficacy of these agents in this regard is impaired by obesity [245,246,247].

On the other side, natural compounds that exhibit anti-oxidant and anti-inflammatory properties have gained medicinal potential for the development of new drugs or as effective co-adjuvant medication in the management of RA [2,33,248,249]. In this line and as a representative of this approach, resveratrol is a polyphenol present in our diet, which it has been widely recognized for its anti-inflammatory, anti-oxidant, anti-cancer and anti-ageing properties [218,248,250,251,252,253]. Previously, we have reported a decreased disease severity in an acute antigen-induced arthritis (AIA) model by dietary oral administration of resveratrol. The reduced arthritis severity was accompanied by significant down-regulation of synovial hyperplasia, as well as by a reduction of local massive infiltration of immune and inflammatory cells. Besides, a potent decrease of cytokine-mediated inflammation and oxidative damage were described [218]. Other research has also demonstrated decreased severity in other animal models of arthritis using resveratrol [254,255,256]. In relation to mitochondria, recent findings of our group has also demonstrated as enhancing autophagic flux and modulating the cross-talk existence with inflammation by limiting inflammasome activation, at least in part, mediate these protective effects [219]. In addition, we have reported that resveratrol can modulate the inflammation induced by mitochondrial dysfunction by decreasing ROS production and NF-κB activation, in normal human synoviocytes [22]. Resveratrol may inhibit NF-κB signaling and inflammation by up-regulating the enzyme adenosine monophosphate kinase (AMPK) and in turns NAD+ and the activity of sirt1 [251]. Additionally, resveratrol as well as other dietary agents could modulate Notch pathway [257,258]. In relation to monocytes and macrophages, several in vitro macrophage studies have also shown that resveratrol pre-treatment can modify the macrophage inflammatory and oxidative response to the inflammatory stimulus and Toll-like receptor (TLR) 4 ligand, the lipopolysaccharide. On the other hand, if resveratrol could lead to a better treatment response in obese RA patients remains to be elucidated. In this regard, several human and animal studies show as resveratrol leads to improved insulin resistance, weight loss, and enhanced glucose homeostasis through promoting fat browning of white adipose tissue by regulating the secretion of adipokines and myokines [259]. In particular, resveratrol alleviates obesity-induced skeletal muscle inflammation via decreasing M1 macrophage polarization and increasing the regulatory T cell population [260]. Similarly, resveratrol reduced obesity in high-fat diet fed mice via modulating the composition and metabolic function of the gut microbiota [261]. In connection with the later, the modulation of intestinal microbiota is a potential therapeutic option for the management of RA [262,263,264]. In fact, gut microbiota is associated with the genotype for RA risk even in the absence of disease [264]. However, all these results were obtained on different models at various time points, with different doses and from different tissues and cell types and it is difficult to extrapolate what is happening in the articular environment. This difficulty is increased by the fact that the effect of resveratrol is organ and tissue-dependent [265]. Finally, some clinical trials evaluate the effect of resveratrol on obesity. However, the heterogeneity in the populations as well as the different administered doses of resveratrol has limited the results obtained [266,267]. Thus, resveratrol studies are controversial and reveal a pleiotropic immunomodulatory property that is dose-time–target cell-dependent [268].

It is worth mentioning that omega-3 fatty acids are other well-known dietary bioactive compounds to have anti-inflammatory properties and beneficial roles in a variety of inflammatory human diseases, including RA. Thus, omega-3 fatty acids are associated with a lower prevalence of anti-cyclic citrullinated peptide autoantibodies in a population at risk for future RA [269]. In relation to macrophages, omega-3 fatty acids suppress both LPS-induced priming and NLRP3 inflammasome activation [270] and also reduce the inflammation by elevating autophagy process [271]. It should be noted that long-term supplementary administration of coenzyme Q10 and omega-3 fatty acids and especially their combination is able to restore the impaired mitochondrial bioenergetics and antioxidant status in a preclinical model of arthritis [272]. In this sense, dietary omega-3 fatty acids have been suggested to counteract insulin resistance development by modulating mitochondrial bioenergetics and ER stress [273].

In the Table 1 are presented a few examples of the most referenced compounds in PubMed over the past five years that demonstrate potential antioxidant and anti-inflammatory benefits. It should be taken into account that the list of natural products that demonstrate potential benefits in the management of RA supported by in vitro and animal studies is too extensive to cover exhaustively, and we direct the readers to other reviews on the topic (for review, see REFs [274,275,276,277,278]).

In the last few years, numerous studies have strongly evidenced that epigenetic signatures are related to nutrients from diet regulates gene expression and in turn are involved in both health and disease [279,280]. In particular, emerging scientific data suggests that resveratrol and other dietary bioactive compounds as well as their gut metabolites could act as epigenetic regulators with a potential impact on human health [281,282,283]. Nevertheless, more studies are needed to evaluate if those natural bioactive agents of a healthy diet protect mitochondria and inhibit the overactivation of mitochondrial oxidative stress and the associated inflammatory response that define RA.

**Table 1 antioxidants-11-01151-t001:** Natural compounds with beneficial effects on oxidative damage and inflammatory response in RA.

Compound	Model	Outcome	References
		Oxidative Markers and Antioxidant Proteins	
**Resveratrol**	AA model/male and female Sprague Dawley rats Human synoviocytes, RA-FLS, RA monocytes, THP-1 cells	Alleviated synovial hyperplasia, inflammatory cell infiltration in synovium and decreased oxidative stress ↑SIRT1 signaling pathway, ↑Nrf2, ↑HO-1, ↑NQO1, ↑miR-29a-3p and miR-23a-3p and NF-κB-p65 inhibition, ↑AMPK, ↓TNF-α, ↓IL-1β, ↓IL-6, ↓HIF-1α ↓mtROS, ↑mitochondrial membrane potential (Δψm), ↓ROS, ↓COX-2, ↓PGE_2_	[22,284,285,286,287,288]
**Curcumin**	CFA arthritic induced male and female Wistar albino rats	Potent antioxidant and suppressor of immune functions of T-cells ↑GSH, ↑GST, ↑GPx, and ↑SOD levels	[289]
**Hesperidin**	AA model/female Wistar rats AA model/male C57BL/6 mice Mononuclear macrophage cell line RAW264.7	Reduce inflammation, improve antioxidant status and modulate apoptotic processes ↓ROS, ↑CAT, ↑GST, i↓NOS ↓TNF-α, ↓INF-γ	[290,291]
**Ferulic acid**	Mononuclear macrophage cell line RAW264.7	↓NF-κB-p65, ↓c-Fos, MMP-9, ↑Nrf2, ↑GSH, ↑CAT, ↑SOD, ROS	[292]
**Quercetin**	AA model/female C57BL/6 mice RA-FLS	Inflammation suppressor and antioxidant defense booster ↓NF-κB-p65, ↓TNF-α, ↓INF-γ, ↓IL-6, ↑IL-4, ↑IL-10 ↑active caspase-3, ↑apoptotic rate, ↓autophagic markers	[293,294,295]
**Gentiopicroside**	AA model/male Sprague Dawley rats CIA model/male C57BL/6J mice RA-FLS	Immunomodulator, analgesic and osteoclastogenesis inhibitor ↓histopathological markers, ↓CD68 ↓NF-κB-p65, ↓TNF-α, ↓IL-1β, I↓L-6, ↓IL-17, ↓VCAM-1, ↓TGF-β, ↓caspase-1, ↑GSH, ↑SOD, ↑GSH-Px	[159,296,297]
**Oleocanthal**	CIA model/male DBA-1/j mice Macrophages	↓NF-κB-p65, ↓IL-1β, ↓INF-γ, ↓IL-6, ↓TNF-α, ↓MMP-3, ↓PGE_2_, ↓iNOS, ↓NO_2_ production, ↑Nrf2, ↑HO-1, ↓NLRP3, ↓active caspase-1, ↓ASC	[298]
**Sulforaphane**	CIA model/male DBA/1J mice PMBCs RAFS	Great ability to induce phase II antioxidant enzymes and exert anti-proliferative effects ↓RANKL, ↓TNF-α, ↓IL-6, ↓IL-17, ↑Nrf2	[157,299]
**Omega 3 fatty acids**	CIA model/female DBA-1 mice CIA model/Fat-1 transgenic mice AA model/male Lewis rats	Anti-inflammatory effect through immune cell inhibition ↓NF-κB-p65, ↓NLRP3, ↓ASC ↓IL-17, I↓L-6, ↓IL-23, ↓ATP, ↓ADP, ↓plasma CoQ9, ↑mitochondrial CoQ9 and CoQ10	[272,300,301]

AA: Adjuvant-induced Arthritis; ADP: Adenosine Diphosphate; AMPK: Adenine Monophosphate Protein Kinase; ASC: Associated Speck-like Protein Containing a CARD; ATP: Adenosine triphosphate; CAT: Catalase; CD68: Cluster of Differentiation 68; coQ: Coenzyme Q; COX-2: Cyclooxygenase-2; GPx: Glutathione Peroxidase; GSH: Glutathione; GST: Glutathione S-transferases; HIF-1α: Hypoxia-inducible Factor-1α; HO-1: Heme Oxygenase 1; IL: Interleukin; INF-γ: Interferon-γ; iNOS: Inducible Nitric Oxide Synthase; NF-κB: Nuclear Factor κB; Nrf2: Nuclear Factor-rythroid-2–Related Factor 2; NLRP3: NLR Family Pyrin Domain Containing 3; NO: Nitric Oxide; NQO1: NAD(P)H quinone dehydrogenase 1; MMP-3: Matrix Metallopeptidase 3; PGE_2:_ Prostaglandin E2; RA: Rheumatoid Arthritis; RA-FLS: Rheumatoid Arthritis Fibroblast-like Synoviocytes; RANKL: Receptor Activator for Nuclear Factor κB Ligand; ROS: Reactive Oxygen Species; SIRT-1: Sirtuin-1; SOD: Superoxide Dismutase; TGF-β: Transforming growth factor-β; THP-1: Tohoku Hospital Pedriatrics-1 Cells; TNF-α: Tumor necrosis factor-α; VCAM-1: Vascular Cell Adhesion Molecule-1.

## 7. Conclusions and Future Perspectives

Undoubtedly, loss of mitochondrial activity can define multiple pathological conditions, including RA (Figure 1) [22,119]. Interestingly, enhanced metabolism is needed to contribute to the abnormal synovial hyperplasia associated with local infiltration of various type of immune and inflammatory cells that define the synovial *pannus*. In this sense, a body of recent research indicates glycolysis rate-limiting enzymes as novel potential regulators of RA pathogenesis [105]. In fact, regulation lactate metabolism imbalance could allow revolutionary pharmacological approaches to restore mitochondrial function in RA [302]. Remarkably, several drugs currently used for the treatment of RA could exert their anti-inflammatory actions by affecting mitochondrial metabolic signaling pathways. In addition, persistent inflammation could lead to some epigenetic marks with negatively influence in the development of RA. In this regard, the impact of nutraceutical components could modulate metabolic inflammation through epigenetic reprogramming. Future studies should elucidate the preservation of mitochondrial activity with natural anti-inflammatory and antioxidant compounds, e.g., resveratrol or omega-3, as a potential strategy for controlling the excessive oxidative stress within mitochondrial inflammatory response in RA. In the same line, the emerging field of redox-medicine [37] aims to controlling excessive oxidative stress between mitochondria to offer both preventive and therapeutic opportunities in pathologies characterized by a plethora of ROS and mitochondrial damage, such as RA. Finally, further and larger studies are needed to better define whether precise metabolic marks could be applying to identify patients with RA according to outcome, disease activity and therapeutic response.

To date, all previous findings support the relevance of mitochondria as new pharmacological targets. The proper maintenance of a good mitochondrial activity could control oxidative stress and successfully represent a new protective and therapeutic new strategy in RA and other inflammatory arthritis. However, in spite of all the advances that have been achieved in the field, the knowledge obtained is still at an early stage and it is necessary more evidence to shed light on how mitochondrial dysfunction could modulate the onset and progression of RA. In this regard, appropriate assay methods to avoid opposite results is the use of not manipulated cells from RA and healthy donors. In addition, most in vitro studies are performed in cultures maintained in a normal atmosphere and with high glucose concentrations, quite different from the hypoxic conditions of rheumatoid joints, which could lead to obtain their energy primarily from anaerobic glycolysis, which probably could mask the obtained results. In the same line, cell–cell crosstalk is a principal contributor to RA pathogenesis. These models should generate more robust and powerful findings. Moreover, most studies evaluate a reduced number of factors when several “omics” studies at different molecular levels, including transcriptome, epigenome, proteomics, and metabolomics should be performed to compare different RA profiles. Many of these approaches require a huge economic and human effort. For this reason, the development of coordinated studies that allow obtaining more solid and robust results is essential to understand the heterogeneity of RA and facilitate the discovery and development of novel drug treatments.

Definitely, it is necessary to keep in mind that prevention is the best treatment. In this sense, a healthy diet based on incorporation of a wide variety of nutrients and preventive nutrition could contribute to reduce the appearance of diseases that arise as a consequence of a deficient or improper diet as several studies suggest could happen with RA [33,249,269,303]. This way requires the participation of different sectors, both public and private, with national policies that promote a healthy diet. In this manner, we can help to fulfill the maxim sought by Hippocrates 25 centuries ago: “Let food be the medicine and medicine be the food”.

## Figures and Tables

**Figure 1 antioxidants-11-01151-f001:**
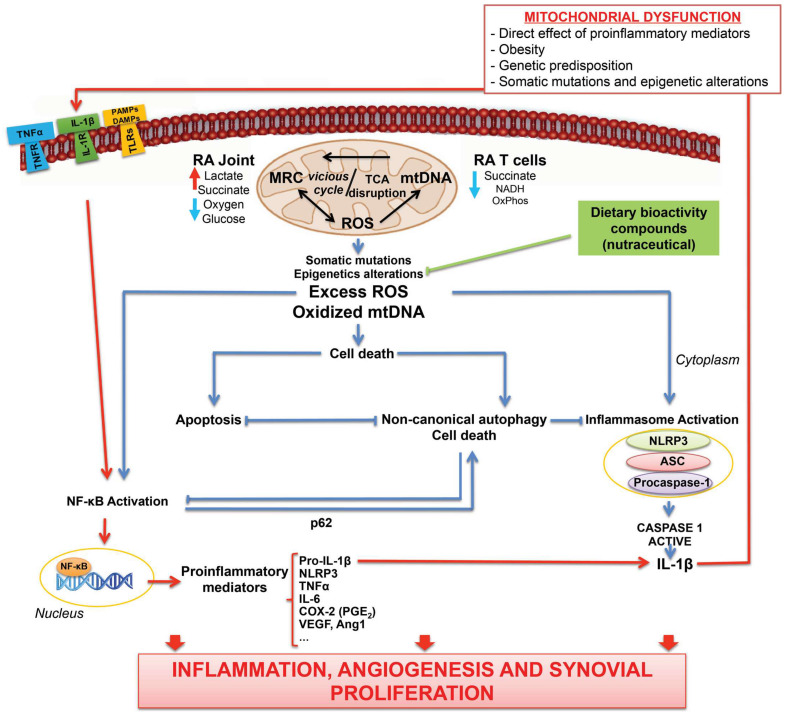
Theoretical model for the multidirectional interplays between mitochondrial oxidative stress, metabolic status, inflammation and cell death in RA. Mitochondria play essential roles at the crossroads of metabolism and innate immunity [54]. Thus, mitochondrial dysfunction derived from several danger signals could activate TCA disruption and thereby favoring a vicious cycle of oxidative/mitochondrial stress. In fact, oxidative damage in synovial tissue is associated with in vivo hypoxic status, high lactate and low glucose levels [3]. Mitochondrial dysfunction can act through modulating innate immunity via redox-sensitive inflammatory pathways (i.e., NF-κB) or direct activation of the inflammasome. Inflammasome activation and NF-κB pathway could work together to activate inflammatory cytokines, thereby leading to overstimulation of the inflammatory response. On the other hand, mitochondria also have a central position in regulating cell death. In this sense, the two main regulated cell death pathways, apoptosis and autophagy, cooperate in a balanced interaction that promotes cell survival or cell death. Additionally, NF-κB could restrict inflammasome activation via elimination of damaged mitochondria through p-62-dependent clearance of damaged mitochondria as well as autophagy modulate NF-κB activation [135]. Additionally, multiple evidence suggests that pathological processes in RA can be shaped by epigenetic mechanisms and that mitochondria are involved in epigenetic regulation [241]. Natural bioactive agents of a healthy diet could protect mitochondria and inhibit the overactivation of mitochondrial oxidative stress and the associated inflammatory response that define RA.

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
