# Peer review of "Mitochondrial Dysfunction and Oxidative Stress in Rheumatoid Arthritis"

_antioxidants, 2022, doi:10.3390/antiox11061151_

Round 1
Reviewer 1 Report
A comprehensive review is presented, about mitochondrial disfunction, oxidative stress metabolic status, inflamatory response, apoptosis and epigenetic events in rheumatoid arthritis (RA). In the title and abstract the authors state that the main purpose of the review is an approach to nutritional interventions (title) and that an adequate nutrition could represent a valuable resource in the prevention of chronic inflamation (abstract). However, these were not the main objectives addressed in this review. Only in the last chapter the efficiency of anti-oxidant treatments was adressed, but only with one exemple: resveratrol. Other studies, with other compounds, should also be explored, once this paper is a review. And I suggest that a comparison between the efficacy of those compounds, and their specific roles in the improvement of mitochondrial function should be discussed.
I consider that the title and the abstract should be changed to be in accordance with the general review that is presented. In the title: the sentence “an approach to nutritional interventions” is not adequate. In the abstract: the sentence “An adequate nutrition could represent a valuable resource in the prevention of chronic inflammation, including RA” is not proved.
Minor changes:
- The word “Introduction” should be written in capital letters, to be in accordance with the other chapters.
- In line 106: delete “has been”
- In line 181: delete “we”; or change “As we commented above” by “As it was commented above”
Author Response
The paper is interesting and well-written. The objectives are clear, the paragraphs are well structured and all the data are correctly presented. The paper should be of interest to scientist working in the field of nutrition, arthritis and oxidative damage, as well as others with closely related research interest. Therefore, this article contain many novelty of interest for the research community and in conclusion, considering the above matters. I retain that this manuscript could merit to be accepted for publication in Antioxidants. However, I have some suggestions to give more value to the manuscript and to attract further interest of the reader.
Thank you very much for your kind comments on our manuscript. We sincerely appreciate your suggestions, which we believe have contributed significantly to improve the understanding and interest of our manuscript.
- A new paragraph should be inserted that takes into account the importance of NOTCHs receptors in mitochondrial dysfunctions since this pathway seems to be very involved in this context and if there are results that also involve nutritional approaches.
Thank you for your interesting suggestion. We included a paragraph and two sentences at the end of the sections 3.1 (Hypoxia, oxidative stress and inflammation), 4.1 (Apoptosis in the pathophysiology of RA) and 6 (Dietary factors on mitochondrial status in RA) respectively, highlighting the role of NOTCHs receptors in mitochondrial dysfunctions and their modulation by bioactive nutrients.
New version:
3.1. Hypoxia, oxidative stress and inflammation
To the best of our knowledge, it has also been reported that hypoxia can induce Notch activation in RA (Chen, et al., 2021). The Notch signalling pathway is a key developmental route that regulates many cellular processes, including proliferation, cell survival/death and differentiation through intracellular signal transmission that involve receptor-ligand interactions between adjacent cells (Zhuang, et al., 2022). In this sense, the activation of Notch signalling are involved in lymphocytes, synoviocytes and endothelial cells of RA patients (Chen, et al., 2021, Gao, et al., 2013, Jiao, et al., 2010, Nakazawa, et al., 2001) being promoted by TNF in RA synoviocytes (Ando, et al., 2003). Also, several studies have shown that Notch activation accelerate the production of proinflammatory mediators in RA (Chen, et al., 2021). On the other hand, Notch-1 and Notch-3 mediate hypoxia-induced synovial fibroblasts activation and angiogenesis in RA (Chen, et al., 2021). In relation to T helper cells from RA patients a significantly altered expression profile of Notch receptors and enhanced activation of Notch signalling is displayed compared with healthy controls (Jiao, et al., 2010). Besides, Notch-regulated miR-223 targets the aryl hydrocarbon receptor pathway and increases cytokine production in macrophages from RA patients (Ogando, et al., 2016). In this regard, it has been described the amelioration of experimental arthritis by silencing miR-223 (Li, et al., 2012) and, in addition,miR-146a modulates macrophage polarization by inhibiting Notch-1 pathway in macrophages (Huang, et al., 2016). Finally, inhibition of Notch signalling ameliorates experimental inflammatory arthritis (Chen, et al., 2021, Jiao, et al., 2014, Park, et al., 2015). Thus, Notch signalling could be a potential pharmacological target for RA treatment.
4.1. Apoptosis in the pathophysiology of RA
…suppresses proliferation and angiogenesis of human fibroblast-like synovial cells in rheumatoid arthritis (Xia, et al., 2018). Also, it has been described as hypoxia reduced RA synovial fibroblast cell apoptosis through Notch-3, whereas an increase in autophagy bodies under hypoxia can be limited by siNotch-3 (Chen, et al., 2021). On the other hand, recent studies identified a new crosstalk between metabolic status…
- DIETARY FACTORS ON MITOCHONDRIAL STATUS IN RA
…Resveratrol may inhibit NF-κB signaling and inflammation by up-regulating the enzyme adenosine monophosphate kinase (AMPK) and in turns NAD+ and the activity of sirt1 (Park, et al., 2012). Also, resveratrol as well as other dietary agents could modulate Notch pathway (Kiesel and Stan, 2022, Truong, et al., 2011). In relation to monocytes and macrophages…
References:
- Ando, K., S. Kanazawa, T. Tetsuka, S. Ohta, X. Jiang, T. Tada, M. Kobayashi, N. Matsui and T. Okamoto. (2003), 'Induction of Notch Signaling by Tumor Necrosis Factor in Rheumatoid Synovial Fibroblasts', Oncogene Vol. 22, No. 49, pp. 7796-803 (included in the new version, reference 87).
- Chen, J., W. Cheng, J. Li, Y. Wang, X. Shen, A. Su, D. Gan, L. Ke, G. Liu, J. Lin, L. Li, X. Bai and P. Zhang. (2021), 'Notch-1 and Notch-3 Mediate Hypoxia-Induced Activation of Synovial Fibroblasts in Rheumatoid Arthritis', Arthritis Rheumatol Vol. 73, No. 10, pp. 1810-1819 (included in the new version, reference 82).
- Chen, J., J. Li, W. Cheng, J. Lin, L. Ke, G. Liu, X. Bai and P. Zhang. (2021), 'Treatment of Collagen-Induced Arthritis Rat Model by Using Notch Signalling Inhibitor', J Orthop Translat Vol. 28, pp. 100-107 (included in the new version, reference 91).
- Gao, W., C. Sweeney, C. Walsh, P. Rooney, J. McCormick, D. J. Veale and U. Fearon. (2013), 'Notch Signalling Pathways Mediate Synovial Angiogenesis in Response to Vascular Endothelial Growth Factor and Angiopoietin 2', Ann Rheum Dis Vol. 72, No. 6, pp. 1080-8 (included in the new version, reference 84).
- Huang, C., X. J. Liu, QunZhou, J. Xie, T. T. Ma, X. M. Meng and J. Li. (2016), 'Mir-146a Modulates Macrophage Polarization by Inhibiting Notch1 Pathway in Raw264.7 Macrophages', Int Immunopharmacol Vol. 32, pp. 46-54 (included in the new version, reference 90).
- Jiao, Z., W. Wang, M. Guo, T. Zhang, L. Chen, Y. Wang, H. You and J. Li. (2010), 'Expression Analysis of Notch-Related Molecules in Peripheral Blood T Helper Cells of Patients with Rheumatoid Arthritis', Scand J Rheumatol Vol. 39, No. 1, pp. 26-32(included in the new version, reference 85).
- Jiao, Z., W. Wang, S. Hua, M. Liu, H. Wang, X. Wang, Y. Chen, H. Xu and L. Lu. (2014), 'Blockade of Notch Signaling Ameliorates Murine Collagen-Induced Arthritis Via Suppressing Th1 and Th17 Cell Responses', Am J Pathol Vol. 184, No. 4, pp. 1085-1093 (included in the new version, reference 93).
- Kiesel, V. A. and S. D. Stan. (2022), 'Modulation of Notch Signaling Pathway by Bioactive Dietary Agents', Int J Mol Sci Vol. 23, No. 7 (included in the new version, reference 258).
- Li, Y. T., S. Y. Chen, C. R. Wang, M. F. Liu, C. C. Lin, I. M. Jou, A. L. Shiau and C. L. Wu. (2012), 'Brief Report: Amelioration of Collagen-Induced Arthritis in Mice by Lentivirus-Mediated Silencing of Microrna-223', Arthritis Rheum Vol. 64, No. 10, pp. 3240-5 (included in the new version, reference 89).
- Nakazawa, M., H. Ishii, H. Aono, M. Takai, T. Honda, S. Aratani, A. Fukamizu, H. Nakamura, S. Yoshino, T. Kobata, K. Nishioka and T. Nakajima. (2001), 'Role of Notch-1 Intracellular Domain in Activation of Rheumatoid Synoviocytes', Arthritis Rheum Vol. 44, No. 7, pp. 1545-54 (included in the new version, reference 86).
- Ogando, J., M. Tardáguila, A. Díaz-Alderete, A. Usategui, V. Miranda-Ramos, D. J. Martínez-Herrera, L. de la Fuente, M. J. García-León, M. C. Moreno, S. Escudero, J. D. Cañete, M. L. Toribio, I. Cases, A. Pascual-Montano, J. L. Pablos and S. Mañes. (2016), 'Notch-Regulated Mir-223 Targets the Aryl Hydrocarbon Receptor Pathway and Increases Cytokine Production in Macrophages from Rheumatoid Arthritis Patients', Sci Rep Vol. 6, pp. 20223 (included in the new version, reference 88).
- Park, J. S., S. H. Kim, K. Kim, C. H. Jin, K. Y. Choi, J. Jang, Y. Choi, A. R. Gwon, S. H. Baik, U. J. Yun, S. Y. Chae, S. Lee, Y. M. Kang, K. C. Lee, T. V. Arumugam, M. P. Mattson, J. H. Park and D. G. Jo. (2015), 'Inhibition of Notch Signalling Ameliorates Experimental Inflammatory Arthritis', Ann Rheum Dis Vol. 74, No. 1, pp. 267-74 (included in the new version, reference 92).
- Park, S. J., F. Ahmad, A. Philp, K. Baar, T. Williams, H. Luo, H. Ke, H. Rehmann, R. Taussig, A. L. Brown, M. K. Kim, M. A. Beaven, A. B. Burgin, V. Manganiello and J. H. Chung. (2012), 'Resveratrol Ameliorates Aging-Related Metabolic Phenotypes by Inhibiting Camp Phosphodiesterases', Cell Vol. 148, No. 3, pp. 421-33 (included in the new version, reference 251).
- Truong, M., M. R. Cook, S. N. Pinchot, M. Kunnimalaiyaan and H. Chen. (2011), 'Resveratrol Induces Notch2-Mediated Apoptosis and Suppression of Neuroendocrine Markers in Medullary Thyroid Cancer', Ann Surg Oncol Vol. 18, No. 5, pp. 1506-11 (included in the new version, reference 257).
- Zhuang, Y., W. Lu, W. Chen, Y. Wu, Q. Wang and Y. Liu. (2022), 'A Narrative Review of the Role of the Notch Signaling Pathway in Rheumatoid Arthritis', Ann Transl Med Vol. 10, No. 6, pp. 371 (included in the new version, reference 83).
- A table showing the nutrients that actually reduced oxidative damage would help the reader to better understand the antioxidant action.
Thank you for your remark. Initially, and as representative of natural compounds that exhibit anti-oxidant and anti-inflammatory properties for the management of RA, we decide to focus on resveratrol and omega-3 fatty acids. The first one for been a polyphenol that has been widely recognized for its anti-inflammatory and anti-oxidant properties, in addition to our extensive experience with this compound. The latter, because the health-promoting effects of n-3 PUFAs are supported by multiple studies in patients. In the new version and following your suggestion, we have included a table (Table I. Natural compounds with beneficial effects on oxidative damage and inflammatory response in RA) showing a few of the most referenced compound in PubMedline over the past five years that demonstrate potential antioxidant and antiinflammatory benefits. The list of natural products that demonstrate potential benefits in the management of RA supported by in vitro and animal studies is too extensive to cover exhaustively, and since this is not the main objective of this review, we decided to include only a few examples. In this regard, we have included in the new version an excellent review published the last year in Antioxidants (Deligiannidou, et al., 2021) along with other publications (Cao, et al., 2020, Dudics, et al., 2018, Gandhi, et al., 2021, Sharma, et al., 2021).
Also, we have included the next paragraph before the last paragraph of 6 section)
New version:
- DIETARY FACTORS ON MITOCHONDRIAL STATUS IN RA
In the table I are presented a few examples of the most referenced compounds in PubMed over the past five years that demonstrate potential antioxidant and antiinflammatory benefits. Should be taken into account that the list of natural products that demonstrate potential benefits in the management of RA supported by in vitro and animal studies is too extensive to cover exhaustively, and we direct the readers to other reviews on the topic (for review, see REFs (Cao, et al., 2020, Deligiannidou, et al., 2021, Dudics, et al., 2018, Gandhi, et al., 2021, Sharma, et al., 2021)).
|
Table I. Natural compounds with beneficial effects on oxidative damage and inflammatory response in RA. |
|||
|
Compound |
Model |
Outcome |
References |
|
|
|
Oxidative Markers and Antioxidant Proteins |
|
|
Resveratrol |
AA model /male and female Sprague–Dawley rats Human synoviocytes, RA-FLS, RA monocytes, THP-1 cells |
Alleviated synovial hyperplasia, inflammatory cell infiltration in synovium and decreased oxidative stress ↑SIRT1 signaling pathway, ↑Nrf2, ↑HO-1, ↑NQO1, ↑miR-29a-3p and miR-23a-3p and NF-κB-p65 inhibition, ↑AMPK, ↓TNF-α, ↓IL-1β, ↓IL-6, ↓HIF-1α ↑mtROS, ↑mitochondrial membrane potential (Δψm), ↓ROS, ↓COX-2, ↓PGE2 |
(Park, et al., 2017, Valcárcel-Ares, et al., 2014, Wang, et al., 2020, Yang, et al., 2018, Zhang, et al., 2016, Zhang, et al., 2019) |
|
Curcumin |
CFA arthritic induced male and female Wistar albino rats |
Potent antioxidant and suppressor of immune functions of T-cells ↑GSH, ↑GST, ↑GPx, and ↑SOD levels |
(Ahmed, et al., 2021) |
|
Hesperidin |
AA model/ female Wistar rats AA model/ male C57BL/6 mice Mononuclear macrophage cell line RAW264.7 |
Reduce inflammation, improve antioxidant status and modulate apoptotic processes ↓ROS, ↑CAT, ↑GST, ↓iNOS ↓TNF-α, ↓INF-γ |
(Adefegha, et al., 2020, Qi, et al., 2019) |
|
Ferulic acid |
Mononuclear macrophage cell line RAW264.7 |
↓NF-κB-p65,↓c-Fos, MMP-9, ↑Nrf2, ↑GSH, ↑CAT, ↑SOD, ↓ROS |
(Doss, et al., 2018) |
|
Quercetin |
AA model /female C57BL/6 mice RA-FLS
|
Inflammation suppressor and antioxidant defense booster ↓NF-κB-p65, ↓TNF-α, ↓INF-γ, ↓IL-6,↑ IL-4, ↑IL-10 ↑active caspase-3, ↑apoptotic rate, ↓autophagic markers |
(Yang, et al., 2018, Yuan, et al., 2020, Zhao, et al., 2020) |
|
Gentiopicroside |
AA model/ male Sprague–Dawley rats CIA model/male C57BL/6J mice RA-FLS |
Immunomodulator, analgesic and osteoclastogenesis inhibitor ↓histopathological markers, ↓CD68 ↓NF-κB-p65, ↓TNF-α, ↓IL-1β, ↓IL-6, ↓IL-17, ↓VCAM-1, ↓TGF-β, ↓caspase-1, ↑GSH, ↑SOD, ↑GSH-Px |
(Jia, et al., 2022, Wang, et al., 2020, Xie, et al., 2019) |
|
Oleocanthal |
CIA model/male DBA-1/j mice Macrophages |
↓NF-κB-p65, ↓IL-1β, ↓INF-γ, ↓IL-6, ↓TNF-α, ↓MMP-3, ↓PGE2, ↓iNOS, ↓NO2 production, ↑Nrf2, ↑HO-1, ↓NLRP3, ↓active caspase-1, ↓ASC |
(Montoya, et al., 2021) |
|
Sulforaphane |
CIA model/male DBA/1J mice PMBCs RAFS |
Great ability to induce phase II antioxidant enzymes and exert anti-proliferative effects ↓RANKL, ↓TNF-α, ↓IL-6, ↓IL-17, ↑Nrf2
|
(Du, et al., 2020, Moon, et al., 2021) |
|
Omega 3 fatty acids |
CIA model/female DBA-1 mice CIA model/Fat-1 transgenic mice AA model /male Lewis rats |
Anti-inflammatory effect through immune cell inhibition ↓NF-κB-p65, ↓NLRP3, ↓ASC ↓IL-17,↓ IL-6, ↓IL-23, ↓ATP, ↓ADP, ↓plasma CoQ9, ↑mitochondrial CoQ9 and CoQ10 |
(Kim, et al., 2018, Kucharská, et al., 2021, Pérez-Martínez, et al., 2020) |
|
AA: Adjuvant-induced Arthritis; ADP: Adenosine Diphosphate; AMPK: Adenine Monophosphate Protein Kinase; ASC: Associated Speck-like Protein Containing a CARD; ATP: Adenosine triphosphate; CAT: Catalase; CD68: Cluster of Differentiation 68; coQ: Coenzyme Q; COX-2: Cyclooxygenase-2; GPx: Glutathione Peroxidase; GSH: Glutathione; GST: Glutathione S-transferases; HIF-1α: Hypoxia-inducible Factor-1α; HO-1: Heme Oxygenase 1; IL: Interleukin; INF-γ: Interferon-γ; iNOS: Inducible Nitric Oxide Synthase; NF-κB: Nuclear Factor κB; Nrf2: Nuclear Factor-rythroid-2–Related Factor 2; NLRP3: NLR Family Pyrin Domain Containing 3; NO: Nitric Oxide; NQO1: NAD(P)H quinone dehydrogenase 1; MMP-3: Matrix Metallopeptidase 3; PGE2: Prostaglandin E2; RA: Rheumatoid Arthritis; RA-FLS: Rheumatoid Arthritis Fibroblast-like Synoviocytes; RANKL: Receptor Activator for Nuclear Factor κB Ligand; ROS: Reactive Oxygen Species; SIRT-1: Sirtuin-1; SOD: Superoxide Dismutase; TGF-β: Transforming growth factor-β; THP-1: Tohoku Hospital Pedriatrics-1 Cells; TNF-α: Tumor necrosis factor-α; VCAM-1: Vascular Cell Adhesion Molecule-1 |
|||
References:
- Adefegha, S. A., N. B. Bottari, D. B. Leal, C. M. de Andrade and M. R. Schetinger. (2020), 'Interferon Gamma/Interleukin-4 Modulation, Anti-Inflammatory and Antioxidant Effects of Hesperidin in Complete Freund's Adjuvant (Cfa)-Induced Arthritis Model of Rats', Immunopharmacol Immunotoxicol Vol. 42, No. 5, pp. 509-520 (included in the new version, reference 290).
- Ahmed, R. H., S. R. Galaly, N. Moustafa, R. R. Ahmed, T. M. Ali, B. H. Elesawy, O. M. Ahmed and M. Abdul-Hamid. (2021), 'Curcumin and Mesenchymal Stem Cells Ameliorate Ankle, Testis, and Ovary Deleterious Histological Changes in Arthritic Rats', Stem Cells Int Vol. 2021, pp. 3516834 (included in the new version, reference 289).
- Cao, F., M. H. Cheng, L. Q. Hu, H. H. Shen, J. H. Tao, X. M. Li, H. F. Pan and J. Gao. (2020), 'Natural Products Action on Pathogenic Cues in Autoimmunity: Efficacy in Systemic Lupus Erythematosus and Rheumatoid Arthritis as Compared to Classical Treatments', Pharmacol Res Vol. 160, pp. 105054 (included in the new version, reference 277).
- Deligiannidou, G. E., V. Gougoula, E. Bezirtzoglou, C. Kontogiorgis and T. K. Constantinides. (2021), 'The Role of Natural Products in Rheumatoid Arthritis: Current Knowledge of Basic in Vitro and in Vivo Research', Antioxidants (Basel) Vol. 10, No. 4 (included in the new version, reference 274).
- Doss, H. M., S. Samarpita, R. Ganesan and M. Rasool. (2018), 'Ferulic Acid, a Dietary Polyphenol Suppresses Osteoclast Differentiation and Bone Erosion Via the Inhibition of Rankl Dependent Nf-Κb Signalling Pathway', Life Sci Vol. 207, pp. 284-295 (included in the new version, reference 292).
- Du, Y., Q. Wang, N. Tian, M. Lu, X. L. Zhang and S. M. Dai. (2020), 'Knockdown of Nrf2 Exacerbates Tnf-', J Immunol Res Vol. 2020, pp. 6670464 (included in the new version, reference 299).
- Dudics, S., D. Langan, R. R. Meka, S. H. Venkatesha, B. M. Berman, C. T. Che and K. D. Moudgil. (2018), 'Natural Products for the Treatment of Autoimmune Arthritis: Their Mechanisms of Action, Targeted Delivery, and Interplay with the Host Microbiome', Int J Mol Sci Vol. 19, No. 9 (included in the new version, reference 276).
- Gandhi, G. R., G. Jothi, T. Mohana, A. B. S. Vasconcelos, M. M. Montalvão, G. Hariharan, G. Sridharan, P. M. Kumar, R. Q. Gurgel, H. B. Li, J. Zhang and R. Y. Gan. (2021), 'Anti-Inflammatory Natural Products as Potential Therapeutic Agents of Rheumatoid Arthritis: A Systematic Review', Phytomedicine Vol. 93, pp. 153766 (included in the new version, reference 275).
- Jia, N., H. Ma, T. Zhang, L. Wang, J. Cui, Y. Zha, Y. Ding and J. Wang. (2022), 'Gentiopicroside Attenuates Collagen-Induced Arthritis in Mice Via Modulating the Cd147/P38/Nf-Κb Pathway', Int Immunopharmacol Vol. 108, pp. 108854 (included in the new version, reference 296).
- Kim, J. Y., K. Lim, K. H. Kim, J. H. Kim, J. S. Choi and S. C. Shim. (2018), 'N-3 Polyunsaturated Fatty Acids Restore Th17 and Treg Balance in Collagen Antibody-Induced Arthritis', PLoS One Vol. 13, No. 3, pp. e0194331 (included in the new version, reference 301).
- Kucharská, J., S. Poništ, O. Vančová, A. Gvozdjáková, O. Uličná, L. Slovák, M. Taghdisiesfejir and K. Bauerová. (2021), 'Treatment with Coenzyme Q10, Omega-3-Polyunsaturated Fatty Acids and Their Combination Improved Bioenergetics and Levels of Coenzyme Q9 and Q10 in Skeletal Muscle Mitochondria in Experimental Model of Arthritis', Physiol Res Vol. 70, No. 5, pp. 723-733 (included in the previous version).
- Montoya, T., M. Sánchez-Hidalgo, M. L. Castejón, M. Rosillo, A. González-Benjumea and C. Alarcón-de-la-Lastra. (2021), 'Dietary Oleocanthal Supplementation Prevents Inflammation and Oxidative Stress in Collagen-Induced Arthritis in Mice', Antioxidants (Basel) Vol. 10, No. 5 (included in the new version, reference 298).
- Moon, S. J., J. Jhun, J. Ryu, J. Y. Kwon, S. Y. Kim, K. Jung, M. L. Cho and J. K. Min. (2021), 'The Anti-Arthritis Effect of Sulforaphane, an Activator of Nrf2, Is Associated with Inhibition of Both B Cell Differentiation and the Production of Inflammatory Cytokines', PLoS One Vol. 16, No. 2, pp. e0245986 (included in the previous version).
- Park, S. Y., S. W. Lee, S. Y. Lee, K. W. Hong, S. S. Bae, K. Kim and C. D. Kim. (2017), 'Sirt1/Adenosine Monophosphate-Activated Protein Kinase Α Signaling Enhances Macrophage Polarization to an Anti-Inflammatory Phenotype in Rheumatoid Arthritis', Front Immunol Vol. 8, pp. 1135 (included in the new version, reference 288).
- Pérez-Martínez, P. I., O. Rojas-Espinosa, V. G. Hernández-Chávez, P. Arce-Paredes and S. Estrada-Parra. (2020), 'Anti-Inflammatory Effect of Omega Unsaturated Fatty Acids and Dialysable Leucocyte Extracts on Collagen-Induced Arthritis in Dba/1 Mice', Int J Exp Pathol Vol. 101, No. 1-2, pp. 55-64 (included in the new version, reference 300).
- Sharma, D., P. Chaubey and V. Suvarna. (2021), 'Role of Natural Products in Alleviation of Rheumatoid Arthritis-a Review', J Food Biochem Vol. 45, No. 4, pp. e13673 (included in the new version, reference 278).
- Qi, W., C. Lin, K. Fan, Z. Chen, L. Liu, X. Feng, H. Zhang, Y. Shao, H. Fang, C. Zhao, R. Zhang and D. Cai. (2019), 'Hesperidin Inhibits Synovial Cell Inflammation and Macrophage Polarization through Suppression of the Pi3k/Akt Pathway in Complete Freund's Adjuvant-Induced Arthritis in Mice', Chem Biol Interact Vol. 306, pp. 19-28 (included in the new version, reference 291).
- Valcárcel-Ares, M. N., R. R. Riveiro-Naveira, C. Vaamonde-García, J. Loureiro, L. Hermida-Carballo, F. J. Blanco and M. J. López-Armada. (2014), 'Mitochondrial Dysfunction Promotes and Aggravates the Inflammatory Response in Normal Human Synoviocytes', Rheumatology (Oxford) Vol. 53, No. 7, pp. 1332-43 (included in the previous version).
- Wang, G., X. Xie, L. Yuan, J. Qiu, W. Duan, B. Xu and X. Chen. (2020), 'Resveratrol Ameliorates Rheumatoid Arthritis Via Activation of Sirt1-Nrf2 Signaling Pathway', Biofactors Vol. 46, No. 3, pp. 441-453(included in the previous version) .
- Wang, M., H. Li, Y. Wang, Y. Hao, Y. Huang, X. Wang, Y. Lu, Y. Du, F. Fu, W. Xin and L. Zhang. (2020), 'Anti-Rheumatic Properties of Gentiopicroside Are Associated with Suppression of Ros-Nf-Κb-Nlrp3 Axis in Fibroblast-Like Synoviocytes and Nf-Κb Pathway in Adjuvant-Induced Arthritis', Front Pharmacol Vol. 11, pp. 515 (included in the previous version).
- Xie, X., H. Li, Y. Wang, Z. Wan, S. Luo, Z. Zhao, J. Liu, X. Wu and X. Li. (2019), 'Therapeutic Effects of Gentiopicroside on Adjuvant-Induced Arthritis by Inhibiting Inflammation and Oxidative Stress in Rats', Int Immunopharmacol Vol. 76, pp. 105840 (included in the new version, reference 297).
- Yang, G., C. C. Chang, Y. Yang, L. Yuan, L. Xu, C. T. Ho and S. Li. (2018), 'Resveratrol Alleviates Rheumatoid Arthritis Via Reducing Ros and Inflammation, Inhibiting Mapk Signaling Pathways, and Suppressing Angiogenesis', J Agric Food Chem (included in the new version, reference 286) .
- Yang, Y., X. Zhang, M. Xu, X. Wu, F. Zhao and C. Zhao. (2018), 'Quercetin Attenuates Collagen-Induced Arthritis by Restoration of Th17/Treg Balance and Activation of Heme Oxygenase 1-Mediated Anti-Inflammatory Effect', Int Immunopharmacol Vol. 54, pp. 153-162 (included in the new version, reference 295).
- Yuan, K., Q. Zhu, Q. Lu, H. Jiang, M. Zhu, X. Li, G. Huang and A. Xu. (2020), 'Quercetin Alleviates Rheumatoid Arthritis by Inhibiting Neutrophil Inflammatory Activities', J Nutr Biochem Vol. 84, pp. 108454 (included in the new version, reference 293).
- Zhang, J., X. Song, W. Cao, J. Lu, X. Wang, G. Wang, Z. Wang and X. Chen. (2016), 'Autophagy and Mitochondrial Dysfunction in Adjuvant-Arthritis Rats Treatment with Resveratrol', Sci Rep Vol. 6, pp. 32928 (included in the new version, reference 285).
- Zhang, Y., G. Wang, T. Wang, W. Cao, L. Zhang and X. Chen. (2019), 'Nrf2-Keap1 Pathway-Mediated Effects of Resveratrol on Oxidative Stress and Apoptosis in Hydrogen Peroxide-Treated Rheumatoid Arthritis Fibroblast-Like Synoviocytes', Ann N Y Acad Sci (included in the new version, reference 287).
- Zhao, J., B. Chen, X. Peng, C. Wang, K. Wang, F. Han and J. Xu. (2020), 'Quercetin Suppresses Migration and Invasion by Targeting Mir-146a/Gata6 Axis in Fibroblast-Like Synoviocytes of Rheumatoid Arthritis', Immunopharmacol Immunotoxicol Vol. 42, No. 3, pp. 221-227 (included in the new version, reference 294).
- It would be advisable to insert a paragraph in the text showing the “Future perspectives” from a nutritional and pharmacological point of view in the light of all these data (different form CONCLUSIONS)
Accordingly to your comments, we have included a new paragraph about future perspectives at the end of the last section. Now the last section (7. CONCLUSIONS) is renamed as (7. CONCLUSIONS AND FUTURE PERSPECTIVES).
New version:
- CONCLUSIONS AND FUTURE PERSPECTIVES
To date, all previous findings support the relevance of mitochondria as new pharmacological targets. The proper maintenance of a good mitochondrial activity could control oxidative stress and successfully represent a new protective and therapeutic new strategy in RA and other inflammatory arthritis. But, in spite of all the advances that have been achieved in the field, the knowledge obtained is still at an early stage and it is necessary more evidence to shed light on how mitochondrial dysfunction could modulate the onset and progression of RA. In this regard, appropriate assay methods to avoid opposite results is the use of not manipulated cells from RA and healthy donors. In addition, most in vitro studies are performed in cultures maintained in a normal atmosphere and with high glucose concentrations, quite different from the hypoxic conditions of rheumatoid joints, which could lead to obtain their energy primarily from anaerobic glycolysis, which probably could mask the obtained results. In the same line, cell–cell crosstalk is a principal contributor to RA pathogenesis. These models should generate more robust and powerful findings. Moreover, most studies evaluate a reduced number of factors when several “omics” studies at different molecular levels, including transcriptome, epigenome, proteomics, and metabolomics should be performed to compare different RA profiles. Many of these approaches require a huge economic and human effort. For this reason, the development of coordinated studies that allow obtaining more solid and robust results is essential to understand the heterogeneity of RA and facilitate the discovery and development of novel drug treatments.
Definitely, it is necessary to keep in mind that prevention is the best treatment. In this sense, healthy diet based on incorporation of a wide variety of nutrients and preventive nutrition could contribute to reduce the appearance of diseases that arise as a consequence of a deficient or improper diet as several studies suggest it could be happen with RA (Coras, et al., 2021, Fatel, et al., 2021, Gan, et al., 2017, Raad, et al., 2021). This way, this requires the participation of different sectors, public and private with national policies that promote a healthy diet. In this manner, we can help to fulfill the maxim sought by Hippocrates 25 centuries ago: “Let food be the medicine and medicine be the food”.
References:
- Coras, R., B. Pedersen, R. Narasimhan, A. Brandy, L. Mateo, A. Prior-Español, A. Kavanaugh, A. M. Armando, M. Jain, O. Quehenberger, M. Martínez-Morillo and M. Guma. (2021), 'Imbalance between Omega-6- and Omega-3-Derived Bioactive Lipids in Arthritis in Older Adults', J Gerontol A Biol Sci Med Sci 76, No. 3, pp. 415-425 (included in the new version, reference 303).
- Fatel, E. C. S., F. T. Rosa, D. F. Alfieri, T. Flauzino, B. M. Scavuzzi, M. A. B. Lozovoy, T. M. V. Iriyoda, A. N. C. Simão and I. Dichi. (2021), 'Beneficial Effects of Fish Oil and Cranberry Juice on Disease Activity and Inflammatory Biomarkers in People with Rheumatoid Arthritis', Nutrition Vol. 86, pp. 111183 (included in the previous version).
- Gan, R. W., M. K. Demoruelle, K. D. Deane, M. H. Weisman, J. H. Buckner, P. K. Gregersen, T. R. Mikuls, J. R. O'Dell, R. M. Keating, T. E. Fingerlin, G. O. Zerbe, M. J. Clare-Salzler, V. M. Holers and J. M. Norris. (2017), 'Omega-3 Fatty Acids Are Associated with a Lower Prevalence of Autoantibodies in Shared Epitope-Positive Subjects at Risk for Rheumatoid Arthritis', Ann Rheum Dis Vol. 76, No. 1, pp. 147-152 (included in the previous text).
- Raad, T., A. Griffin, E. S. George, L. Larkin, A. Fraser, N. Kennedy and A. C. Tierney. (2021), 'Dietary Interventions with or without Omega-3 Supplementation for the Management of Rheumatoid Arthritis: A Systematic Review', Nutrients Vol. 13, No. 10 (included in the new version, reference 249).
- A diagram showing the link between mitochondrial dysfunction and the onset of arthritic damage, with all mediators involved
We are not sure if we have understood this suggestion correctly. Figure 1 of the previous version defines a theoretical model for the multidirectional interplays between mitochondrial oxidative stress, metabolic status, and inflammation and cell death in RA, with most of the mediators/pathways described in this review included. Now, we have developed a straightforward diagram showing the possible link between mitochondrial dysfunction and the onset of arthritic damage. We hope the new diagram will adequately address your concern.

Reviewer 2 Report
Referee’s comment
Article n° Antioxidants-1732705
Title: Mitochondrial dysfunction and oxidative stress in rheumatoid arthritis: an approach to nutritional interventions
Authors: López-Armada MJ, Fernández-RodríguezJA, Blanco FJ.
The paper is interesting and well-written. The objectives are clear, the paragraphs are well structured and all the data are correctly presented. The paper should be of interest to scientist working in the field of nutrition, arthritis and oxidative damage, as well as other s with closely related research interest. Therefore, this article contain many novelty of interest for the research community and in conclusion, considering the above matters, I retain that this manuscript could merit to be accepted for publication in Antioxidants. However I have some suggestions to give more value to the manuscript and to attract further interest of the reader.
1) A new paragraph should be inserted that takes into account the importance of NOTCHs receptors in mitochondrial dysfunctions since this pathway seems to be very involved in this context, and if there are results that also involve nutritional approaches.
2) A table showing the nutrients that actually reduced oxidative damage would help the reader to better understand the antioxidant action.
3) It would be advisable to insert a paragraph in the text showing the “Future perspectives” from a nutritional and pharmacological point of view in the light of all these data (different from CONCLUSIONS).
4) A diagram showing the link between mitochondrial dysfunction and the onset of arthritic damage, with all mediators involved.
Author Response
RESPONSE to REVIEWER 2 COMMENTS
A comprehensive review is presented, about mitochondrial dysfunction, oxidative stress metabolic status, inflammatory response, apoptosis and epigenetic events in rheumatoid arthritis (RA). In the title and abstract the authors state that the main purpose of the review is an approach to nutritional interventions (title) and that an adequate nutrition could represent a valuable resource in the prevention of chronic inflammation (abstract). However, these were not the main objectives addressed in this review. Only in the last chapter the efficiency of anti-oxidant treatments was addressed, but only with one example: resveratrol. Other studies, with other compounds, should also be explored, once this paper is a review. And I suggest that a comparison between the efficacy of this compounds, and their specific roles in the improvement of mitochondrial function should be discussed.
First of all, thank you for taking the time to evaluate our work. We sincerely appreciate your suggestions, which we believe have helped considerably to improve the understanding of our manuscript. We hope that the following comments will adequately answer to each of the points that you have made to us.
I consider that the title and the abstract should be changed to be in accordance with the general review that is presented. In the title: the sentence “an approach to nutritional interventions” is not adequate. In the abstract: the sentence “An adequate nutrition could represent a valuable resource in the prevention of chronic inflammation, including RA” is not proved.
According with your suggestion, we have modified the title and the abstract of the review by a more specific heading and summary that fits better with the topics of the current review.
New version (Title):
“Mitochondrial dysfunction and oxidative stress in rheumatoid arthritis”
New version (Abstract):
Control of excessive mitochondrial oxidative stress could provide new targets for both preventive and therapeutic interventions in the treatment of chronic inflammation or any pathology that develops under an inflammatory scenario, such as rheumatoid arthritis (RA). Increasing evidence has demonstrated the role of mitochondrial alterations in autoimmune diseases mainly due to the interplay between metabolism and innate immunity, but also in the modulation of inflammatory response of resident cells, such as synoviocytes. Thus, mitochondrial dysfunction derived from several danger signals could activate tricarboxylic acid (TCA) disruption and thereby favoring a vicious cycle of oxidative/mitochondrial stress. Mitochondrial dysfunction can act through modulating innate immunity via redox-sensitive inflammatory pathways or direct activation of the inflammasome. Besides, mitochondria also have a central role in regulating cell death, which is deeply altered in RA. Additionally, multiple evidence suggests that pathological processes in RA can be shaped by epigenetic mechanisms and that in turn, mitochondria are involved in epigenetic regulation. Finally, we will discuss about the involvement of some dietary components in the onset and progression of RA.
Previous version (Abstract):
Control of excessive mitochondrial oxidative stress could provide new therapeutic targets for both preventive and therapeutic interventions in the treatment of chronic inflammation or any pathology that develops along with inflammation, such as rheumatoid arthritis (RA). Increasing evidence has demonstrated the role of mitochondrial alterations in autoimmune diseases mainly due to their active participation in both the innate and adaptive immune systems, but also in the modulation of inflammatory response of resident cells, such as synoviocytes. Although it is clear that diet is gaining more prominence in the context of a healthy lifestyle, there is still a long way to go. An adequate nutrition could represent a valuable resource in the prevention of chronic inflammation, including RA.
Other studies, with other compounds, should also be explored, once this paper is a review.
Thank you for your remark. Initially, and as representative of natural compounds that exhibit anti-oxidant and anti-inflammatory properties for the management of RA, we decide to focus on resveratrol and omega-3 fatty acids. The first one, for been a polyphenol that has been widely recognized for its anti-inflammatory and anti-oxidant properties, in addition to our extensive experience with this compound. The latter, because the health-promoting effects of n-3 PUFAs are supported by multiple studies in patients. In the new version and following your suggestion, we have included a table (Table I. Natural compounds with beneficial effects on oxidative damage and inflammatory response in RA) showing a few of the most referenced compound in PubMed over the past five years that demonstrate potential antioxidant and antiinflammatory benefits. The list of natural products that demonstrate potential benefits in the management of RA supported by in vitro and animal studies is too extensive to cover exhaustively, and since this is not the main objective of this review, we decided to include only a few examples. In this regard, we have included in the new version an excellent review published the last year in Antioxidants (Deligiannidou, et al., 2021) along with other publications (Cao, et al., 2020, Dudics, et al., 2018, Gandhi, et al., 2021, Sharma, et al., 2021).
Also, we have included the next paragraph before the last paragraph of 6 section)
New version:
- DIETARY FACTORS ON MITOCHONDRIAL STATUS IN RA
In the table I are presented a few examples of the most referenced compounds in PubMed over the past five years that demonstrate potential antioxidant and antiinflammatory benefits. Should be taken into account that the list of natural products that demonstrate potential benefits in the management of RA supported by in vitro and animal studies is too extensive to cover exhaustively, and we direct the readers to other reviews on the topic (for review, see REFs (Cao, et al., 2020, Deligiannidou, et al., 2021, Dudics, et al., 2018, Gandhi, et al., 2021, Sharma, et al., 2021)).
|
Table I. Natural compounds with beneficial effects on oxidative damage and inflammatory response in RA. |
|||
|
Compound |
Model |
Outcome |
References |
|
|
|
Oxidative Markers and Antioxidant Proteins |
|
|
Resveratrol |
AA model /male and female Sprague–Dawley rats Human synoviocytes, RA-FLS, RA monocytes, THP-1 cells |
Alleviated synovial hyperplasia, inflammatory cell infiltration in synovium and decreased oxidative stress ↑SIRT1 signaling pathway, ↑Nrf2, ↑HO-1, ↑NQO1, ↑miR-29a-3p and miR-23a-3p and NF-κB-p65 inhibition, ↑AMPK, ↓TNF-α, ↓IL-1β, ↓IL-6, ↓HIF-1α ↑mtROS, ↑mitochondrial membrane potential (Δψm), ↓ROS, ↓COX-2, ↓PGE2 |
(Park, et al., 2017, Valcárcel-Ares, et al., 2014, Wang, et al., 2020, Yang, et al., 2018, Zhang, et al., 2016, Zhang, et al., 2019) |
|
Curcumin |
CFA arthritic induced male and female Wistar albino rats |
Potent antioxidant and suppressor of immune functions of T-cells ↑GSH, ↑GST, ↑GPx, and ↑SOD levels |
(Ahmed, et al., 2021) |
|
Hesperidin |
AA model/ female Wistar rats AA model/ male C57BL/6 mice Mononuclear macrophage cell line RAW264.7 |
Reduce inflammation, improve antioxidant status and modulate apoptotic processes ↓ROS, ↑CAT, ↑GST, ↓iNOS ↓TNF-α, ↓INF-γ |
(Adefegha, et al., 2020, Qi, et al., 2019) |
|
Ferulic acid |
Mononuclear macrophage cell line RAW264.7 |
↓NF-κB-p65,↓c-Fos, MMP-9, ↑Nrf2, ↑GSH, ↑CAT, ↑SOD, ↓ROS |
(Doss, et al., 2018) |
|
Quercetin |
AA model /female C57BL/6 mice RA-FLS
|
Inflammation suppressor and antioxidant defense booster ↓NF-κB-p65, ↓TNF-α, ↓INF-γ, ↓IL-6,↑ IL-4, ↑IL-10 ↑active caspase-3, ↑apoptotic rate, ↓autophagic markers |
(Yang, et al., 2018, Yuan, et al., 2020, Zhao, et al., 2020) |
|
Gentiopicroside |
AA model/ male Sprague–Dawley rats CIA model/male C57BL/6J mice RA-FLS |
Immunomodulator, analgesic and osteoclastogenesis inhibitor ↓histopathological markers, ↓CD68 ↓NF-κB-p65, ↓TNF-α, ↓IL-1β, ↓IL-6, ↓IL-17, ↓VCAM-1, ↓TGF-β, ↓caspase-1, ↑GSH, ↑SOD, ↑GSH-Px |
(Jia, et al., 2022, Wang, et al., 2020, Xie, et al., 2019) |
|
Oleocanthal |
CIA model/male DBA-1/j mice Macrophages |
↓NF-κB-p65, ↓IL-1β, ↓INF-γ, ↓IL-6, ↓TNF-α, ↓MMP-3, ↓PGE2, ↓iNOS, ↓NO2 production, ↑Nrf2, ↑HO-1, ↓NLRP3, ↓active caspase-1, ↓ASC |
(Montoya, et al., 2021) |
|
Sulforaphane |
CIA model/male DBA/1J mice PMBCs RAFS |
Great ability to induce phase II antioxidant enzymes and exert anti-proliferative effects ↓RANKL, ↓TNF-α, ↓IL-6, ↓IL-17, ↑Nrf2
|
(Du, et al., 2020, Moon, et al., 2021) |
|
Omega 3 fatty acids |
CIA model/female DBA-1 mice CIA model/Fat-1 transgenic mice AA model /male Lewis rats |
Anti-inflammatory effect through immune cell inhibition ↓NF-κB-p65, ↓NLRP3, ↓ASC ↓IL-17,↓ IL-6, ↓IL-23, ↓ATP, ↓ADP, ↓plasma CoQ9, ↑mitochondrial CoQ9 and CoQ10 |
(Kim, et al., 2018, Kucharská, et al., 2021, Pérez-Martínez, et al., 2020) |
|
AA: Adjuvant-induced Arthritis; ADP: Adenosine Diphosphate; AMPK: Adenine Monophosphate Protein Kinase; ASC: Associated Speck-like Protein Containing a CARD; ATP: Adenosine triphosphate; CAT: Catalase; CD68: Cluster of Differentiation 68; coQ: Coenzyme Q; COX-2: Cyclooxygenase-2; GPx: Glutathione Peroxidase; GSH: Glutathione; GST: Glutathione S-transferases; HIF-1α: Hypoxia-inducible Factor-1α; HO-1: Heme Oxygenase 1; IL: Interleukin; INF-γ: Interferon-γ; iNOS: Inducible Nitric Oxide Synthase; NF-κB: Nuclear Factor κB; Nrf2: Nuclear Factor-rythroid-2–Related Factor 2; NLRP3: NLR Family Pyrin Domain Containing 3; NO: Nitric Oxide; NQO1: NAD(P)H quinone dehydrogenase 1; MMP-3: Matrix Metallopeptidase 3; PGE2: Prostaglandin E2; RA: Rheumatoid Arthritis; RA-FLS: Rheumatoid Arthritis Fibroblast-like Synoviocytes; RANKL: Receptor Activator for Nuclear Factor κB Ligand; ROS: Reactive Oxygen Species; SIRT-1: Sirtuin-1; SOD: Superoxide Dismutase; TGF-β: Transforming growth factor-β; THP-1: Tohoku Hospital Pedriatrics-1 Cells; TNF-α: Tumor necrosis factor-α; VCAM-1: Vascular Cell Adhesion Molecule-1 |
|||
References:
- Adefegha, S. A., N. B. Bottari, D. B. Leal, C. M. de Andrade and M. R. Schetinger. (2020), 'Interferon Gamma/Interleukin-4 Modulation, Anti-Inflammatory and Antioxidant Effects of Hesperidin in Complete Freund's Adjuvant (Cfa)-Induced Arthritis Model of Rats', Immunopharmacol Immunotoxicol Vol. 42, No. 5, pp. 509-520 (included in the new version, reference 290).
- Ahmed, R. H., S. R. Galaly, N. Moustafa, R. R. Ahmed, T. M. Ali, B. H. Elesawy, O. M. Ahmed and M. Abdul-Hamid. (2021), 'Curcumin and Mesenchymal Stem Cells Ameliorate Ankle, Testis, and Ovary Deleterious Histological Changes in Arthritic Rats', Stem Cells Int Vol. 2021, pp. 3516834 (included in the new version, reference 289).
- Cao, F., M. H. Cheng, L. Q. Hu, H. H. Shen, J. H. Tao, X. M. Li, H. F. Pan and J. Gao. (2020), 'Natural Products Action on Pathogenic Cues in Autoimmunity: Efficacy in Systemic Lupus Erythematosus and Rheumatoid Arthritis as Compared to Classical Treatments', Pharmacol Res Vol. 160, pp. 105054 (included in the new version, reference 277).
- Deligiannidou, G. E., V. Gougoula, E. Bezirtzoglou, C. Kontogiorgis and T. K. Constantinides. (2021), 'The Role of Natural Products in Rheumatoid Arthritis: Current Knowledge of Basic in Vitro and in Vivo Research', Antioxidants (Basel) Vol. 10, No. 4 (included in the new version, reference 274).
- Doss, H. M., S. Samarpita, R. Ganesan and M. Rasool. (2018), 'Ferulic Acid, a Dietary Polyphenol Suppresses Osteoclast Differentiation and Bone Erosion Via the Inhibition of Rankl Dependent Nf-Κb Signalling Pathway', Life Sci Vol. 207, pp. 284-295 (included in the new version, reference 292).
- Du, Y., Q. Wang, N. Tian, M. Lu, X. L. Zhang and S. M. Dai. (2020), 'Knockdown of Nrf2 Exacerbates Tnf-', J Immunol Res Vol. 2020, pp. 6670464 (included in the new version, reference 299).
- Dudics, S., D. Langan, R. R. Meka, S. H. Venkatesha, B. M. Berman, C. T. Che and K. D. Moudgil. (2018), 'Natural Products for the Treatment of Autoimmune Arthritis: Their Mechanisms of Action, Targeted Delivery, and Interplay with the Host Microbiome', Int J Mol Sci Vol. 19, No. 9 (included in the new version, reference 276).
- Gandhi, G. R., G. Jothi, T. Mohana, A. B. S. Vasconcelos, M. M. Montalvão, G. Hariharan, G. Sridharan, P. M. Kumar, R. Q. Gurgel, H. B. Li, J. Zhang and R. Y. Gan. (2021), 'Anti-Inflammatory Natural Products as Potential Therapeutic Agents of Rheumatoid Arthritis: A Systematic Review', Phytomedicine Vol. 93, pp. 153766 (included in the new version, reference 275).
- Jia, N., H. Ma, T. Zhang, L. Wang, J. Cui, Y. Zha, Y. Ding and J. Wang. (2022), 'Gentiopicroside Attenuates Collagen-Induced Arthritis in Mice Via Modulating the Cd147/P38/Nf-Κb Pathway', Int Immunopharmacol Vol. 108, pp. 108854 (included in the new version, reference 296).
- Kim, J. Y., K. Lim, K. H. Kim, J. H. Kim, J. S. Choi and S. C. Shim. (2018), 'N-3 Polyunsaturated Fatty Acids Restore Th17 and Treg Balance in Collagen Antibody-Induced Arthritis', PLoS One Vol. 13, No. 3, pp. e0194331 (included in the new version, reference 301).
- Kucharská, J., S. Poništ, O. Vančová, A. Gvozdjáková, O. Uličná, L. Slovák, M. Taghdisiesfejir and K. Bauerová. (2021), 'Treatment with Coenzyme Q10, Omega-3-Polyunsaturated Fatty Acids and Their Combination Improved Bioenergetics and Levels of Coenzyme Q9 and Q10 in Skeletal Muscle Mitochondria in Experimental Model of Arthritis', Physiol Res Vol. 70, No. 5, pp. 723-733 (included in the previous version).
- Montoya, T., M. Sánchez-Hidalgo, M. L. Castejón, M. Rosillo, A. González-Benjumea and C. Alarcón-de-la-Lastra. (2021), 'Dietary Oleocanthal Supplementation Prevents Inflammation and Oxidative Stress in Collagen-Induced Arthritis in Mice', Antioxidants (Basel) Vol. 10, No. 5 (included in the new version, reference 298).
- Moon, S. J., J. Jhun, J. Ryu, J. Y. Kwon, S. Y. Kim, K. Jung, M. L. Cho and J. K. Min. (2021), 'The Anti-Arthritis Effect of Sulforaphane, an Activator of Nrf2, Is Associated with Inhibition of Both B Cell Differentiation and the Production of Inflammatory Cytokines', PLoS One Vol. 16, No. 2, pp. e0245986 (included in the previous version).
- Park, S. Y., S. W. Lee, S. Y. Lee, K. W. Hong, S. S. Bae, K. Kim and C. D. Kim. (2017), 'Sirt1/Adenosine Monophosphate-Activated Protein Kinase Α Signaling Enhances Macrophage Polarization to an Anti-Inflammatory Phenotype in Rheumatoid Arthritis', Front Immunol Vol. 8, pp. 1135 (included in the new version, reference 288).
- Pérez-Martínez, P. I., O. Rojas-Espinosa, V. G. Hernández-Chávez, P. Arce-Paredes and S. Estrada-Parra. (2020), 'Anti-Inflammatory Effect of Omega Unsaturated Fatty Acids and Dialysable Leucocyte Extracts on Collagen-Induced Arthritis in Dba/1 Mice', Int J Exp Pathol Vol. 101, No. 1-2, pp. 55-64 (included in the new version, reference 300).
- Sharma, D., P. Chaubey and V. Suvarna. (2021), 'Role of Natural Products in Alleviation of Rheumatoid Arthritis-a Review', J Food Biochem Vol. 45, No. 4, pp. e13673 (included in the new version, reference 278).
- Qi, W., C. Lin, K. Fan, Z. Chen, L. Liu, X. Feng, H. Zhang, Y. Shao, H. Fang, C. Zhao, R. Zhang and D. Cai. (2019), 'Hesperidin Inhibits Synovial Cell Inflammation and Macrophage Polarization through Suppression of the Pi3k/Akt Pathway in Complete Freund's Adjuvant-Induced Arthritis in Mice', Chem Biol Interact Vol. 306, pp. 19-28 (included in the new version, reference 291).
- Valcárcel-Ares, M. N., R. R. Riveiro-Naveira, C. Vaamonde-García, J. Loureiro, L. Hermida-Carballo, F. J. Blanco and M. J. López-Armada. (2014), 'Mitochondrial Dysfunction Promotes and Aggravates the Inflammatory Response in Normal Human Synoviocytes', Rheumatology (Oxford) Vol. 53, No. 7, pp. 1332-43 (included in the previous version).
- Wang, G., X. Xie, L. Yuan, J. Qiu, W. Duan, B. Xu and X. Chen. (2020), 'Resveratrol Ameliorates Rheumatoid Arthritis Via Activation of Sirt1-Nrf2 Signaling Pathway', Biofactors Vol. 46, No. 3, pp. 441-453(included in the previous version) .
- Wang, M., H. Li, Y. Wang, Y. Hao, Y. Huang, X. Wang, Y. Lu, Y. Du, F. Fu, W. Xin and L. Zhang. (2020), 'Anti-Rheumatic Properties of Gentiopicroside Are Associated with Suppression of Ros-Nf-Κb-Nlrp3 Axis in Fibroblast-Like Synoviocytes and Nf-Κb Pathway in Adjuvant-Induced Arthritis', Front Pharmacol Vol. 11, pp. 515 (included in the previous version).
- Xie, X., H. Li, Y. Wang, Z. Wan, S. Luo, Z. Zhao, J. Liu, X. Wu and X. Li. (2019), 'Therapeutic Effects of Gentiopicroside on Adjuvant-Induced Arthritis by Inhibiting Inflammation and Oxidative Stress in Rats', Int Immunopharmacol Vol. 76, pp. 105840 (included in the new version, reference 297).
- Yang, G., C. C. Chang, Y. Yang, L. Yuan, L. Xu, C. T. Ho and S. Li. (2018), 'Resveratrol Alleviates Rheumatoid Arthritis Via Reducing Ros and Inflammation, Inhibiting Mapk Signaling Pathways, and Suppressing Angiogenesis', J Agric Food Chem (included in the new version, reference 286) .
- Yang, Y., X. Zhang, M. Xu, X. Wu, F. Zhao and C. Zhao. (2018), 'Quercetin Attenuates Collagen-Induced Arthritis by Restoration of Th17/Treg Balance and Activation of Heme Oxygenase 1-Mediated Anti-Inflammatory Effect', Int Immunopharmacol Vol. 54, pp. 153-162 (included in the new version, reference 295).
- Yuan, K., Q. Zhu, Q. Lu, H. Jiang, M. Zhu, X. Li, G. Huang and A. Xu. (2020), 'Quercetin Alleviates Rheumatoid Arthritis by Inhibiting Neutrophil Inflammatory Activities', J Nutr Biochem Vol. 84, pp. 108454 (included in the new version, reference 293).
- Zhang, J., X. Song, W. Cao, J. Lu, X. Wang, G. Wang, Z. Wang and X. Chen. (2016), 'Autophagy and Mitochondrial Dysfunction in Adjuvant-Arthritis Rats Treatment with Resveratrol', Sci Rep Vol. 6, pp. 32928 (included in the new version, reference 285).
- Zhang, Y., G. Wang, T. Wang, W. Cao, L. Zhang and X. Chen. (2019), 'Nrf2-Keap1 Pathway-Mediated Effects of Resveratrol on Oxidative Stress and Apoptosis in Hydrogen Peroxide-Treated Rheumatoid Arthritis Fibroblast-Like Synoviocytes', Ann N Y Acad Sci (included in the new version, reference 287).
- Zhao, J., B. Chen, X. Peng, C. Wang, K. Wang, F. Han and J. Xu. (2020), 'Quercetin Suppresses Migration and Invasion by Targeting Mir-146a/Gata6 Axis in Fibroblast-Like Synoviocytes of Rheumatoid Arthritis', Immunopharmacol Immunotoxicol Vol. 42, No. 3, pp. 221-227 (included in the new version, reference 294).
And I suggest that a comparison between the efficacy of this compounds, and their specific roles in the improvement of mitochondrial function should be discussed.
We completely agree with your comments on this point. Indeed, we are actually tackling in vitro analyses on the human monocyte cell line THP-1 with different natural products (resveratrol, sulforaphane and curcumin) evaluating mitochondrial function (mtROS, Δψm, basal and maximal respiration by standard Seahorse Mito Stress Test), baseline and after stimulation with mitoDAMPs. Even, it is necessary to create more effective research methods to clarify the mechanism of action of different natural products or isolated compounds in the treatment of RA, and to shed light on whether those natural bioactive agents could protect mitochondria and inhibit the overactivation of mitochondrial oxidative stress and associated inflammatory response that characterize arthritic disease.
Minor changes:
- The world “Introduction” should be written in capital letters, to be in accordance with the other chapters.
- In line 106: delate “has been”
- In line 181: delate we; or change “As we commented above” by “As it was commented above”
We thank you for detecting these grammar and typographical errors. We have corrected all of them. The introduced changes can be tracked in the marked version of the manuscript.
